# Chemoproteomic discovery of a human RNA ligase

Yizhi Yuan [1,2], Florian M. Stumpf[1,2], Lisa A. Schlor[1,2], Olivia P. Schmidt[1], Philip Saumer [1,2], Luisa B. Huber[1,2], Matthias Frese[1,2], Eva Höllmüller[1,2], Martin Scheffner [2,3], Florian Stengel [2,3], Kay Diederichs [2,3] & Andreas Marx [1,2] ✉

RNA ligases are present across all forms of life. While enzymatic RNA ligation between 5′-PO₄ and 3′-OH termini is prevalent in viruses, fungi, and plants, such RNA ligases are yet to be identified in vertebrates. Here, using a nucleotide-based chemical probe targeting human AMPylated proteome, we have enriched and identified the hitherto uncharacterised human protein chromosome 12 open reading frame 29 (C12orf29) as a human enzyme promoting RNA ligation between 5′-PO₄ and 3′-OH termini. C12orf29 catalyses ATP-dependent RNA ligation via a three-step mechanism, involving tandem auto- and RNA AMPylation. Knock-out of *C12ORF29* gene impedes the cellular resilience to oxidative stress featuring concurrent RNA degradation, which suggests a role of C12orf29 in maintaining RNA integrity. These data provide the groundwork for establishing a human RNA repair pathway.

RNA ligases play vital roles in sealing RNA strands during intron-containing tRNA splicing[1], tRNA repair[2], mRNA splicing in the unfolded-protein response (UPR)[3], RNA recombination[4], as well as biogenesis of circular RNAs[5]. Across all life forms, proteinaceous RNA ligases are present with distinct catalytic mechanisms[1]. Like DNA ligases, RNA ligases are known that join 5′-PO₄ and 3′-OH termini of RNA via a classic three-step mechanism[1] (Fig. 1a). First, the RNA ligase undergoes adenosine 5′-triphosphate (ATP)-dependent auto-AMPylation (also known as auto-adenylylation) at the catalytic lysine residue. The AMP is subsequently transferred from the ligase-(lysyl-*N*)-AMP to the 5′-PO₄ end of RNA (pRNA) to yield the RNA-adenylate intermediate (AppRNA). Finally, the two RNA ends are ligated by a phosphodiester bond upon nucleophilic attack of the 3′-OH to the AppRNA, liberating the AMP. These enzymes, termed 5′−3′ RNA ligases[1], have been heavily exploited as molecular biology tools in RNA editing and sequencing[6,7]. Although the existence of a 5′−3′ RNA ligase in HeLa cells has been suggested earlier[8], such an enzyme remains to be identified in vertebrates. So far, only one proteinaceous RNA ligase has been identified in human, which is a guanidine 5′-triphosphate

(GTP)-dependent 3′-5′ RNA ligase that joins 2′,3′-cyclic PO₄ (cPO₄) and 5′-OH termini of RNA as a subunit of a human tRNA splicing ligase complex[9].

We are interested in nucleotides and their involvement in processes of post-translational modification (PTM)[10,11]. AMPylation is a PTM in which an AMP molecule is covalently bound to a side chain or the C-terminus of a protein[12,13], which is often catalysed by an enzyme using ATP as a co-substrate. Over the past decade, AMPylation has been found to participate in versatile biological processes, such as pathogen infection[14–17], UPR[18,19], and cellular redox homoeostasis[20]. Several ATP derivatives have been designed for profiling AMPylation[20–22]. In recent studies masked AMP analogues were used as probes to investigate AMPylation[23,24]. After internalisation they were processed to ATP and used by the cellular machinery for protein AMPylation.

Here, to further dissect the human AMPylation proteome, we have synthesised an alkyne-modified chemical probe based on diadenosine triphosphate (Ap₃A, Fig. 1b)—a naturally occurring nucleotide whose concentration increases as a result of cellular stress and with higher

[1]Department of Chemistry, University of Konstanz, Universitätsstraße 10, 78457 Konstanz, Germany. [2]Konstanz Research School Chemical Biology, University of Konstanz, Universitätsstraße 10, 78457 Konstanz, Germany. [3]Department of Biology, University of Konstanz, Universitätsstraße 10, 78457 Konstanz, Germany. ✉e-mail: andreas.marx@uni-konstanz.de

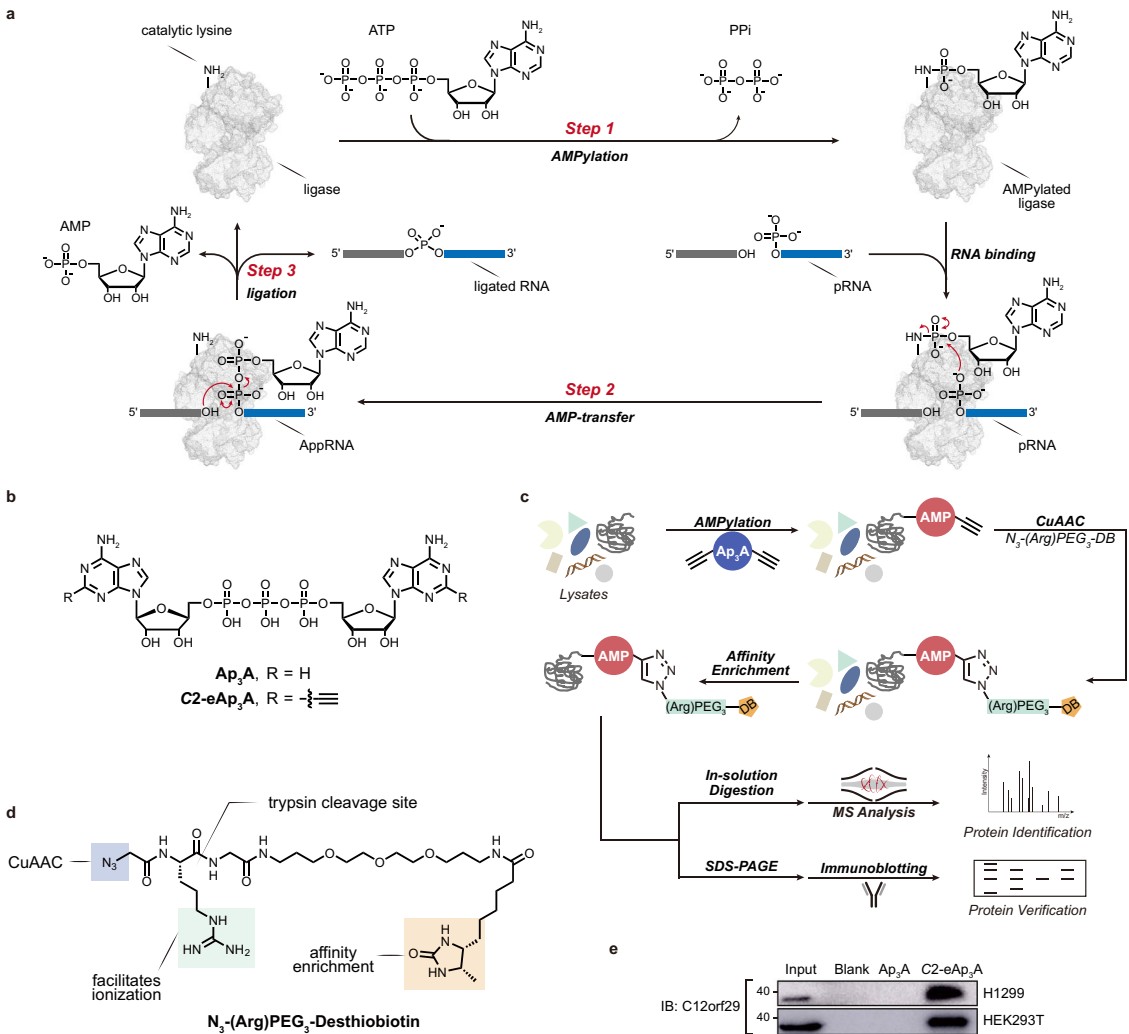

**Fig. 1 | RNA ligase mechanism and identification of C12orf29 by chemical pro-teomics using modified Ap₃A probe. a** Schematic display of the three-step mechanism of RNA ligation by a 5′–3′ RNA ligase. In step 1, the ligase is auto-AMPylated on the catalytic lysine using ATP as the co-substrate. In step 2, the AMP is transferred from the catalytic lysine to the 5′-PO₄ end of RNA (pRNA), giving the RNA-adenylate intermediate (AppRNA). In step 3, the ligated RNA is obtained upon the attack of 3′-OH to the AppRNA in the presence of the ligase along with the liberation of AMP. PPi, pyrophosphate. **b** Structures of Ap₃A analogues employed in this study. **c** Schematic display of the workflow for the identification of C12orf29. Cell lysates are incubated with *C*2-eAp₃A or controls. AMPylated proteins are expected to bear ethynyl functionalities that enable selective modification with an affinity tag desthiobiotin (DB) via CuAAC. Labelled proteins are enriched and identified by ABPP, and further verified by immunoblotting. **d** Structure of the azide-bearing desthiobiotin as affinity tag. **e** Affinity enrichment of C12orf29 from two cell lysates verified by immunoblotting (representative images of *n* = 3). Source data are provided as a Source Data file.

stability in cell lysates than ATP[25]. Using this chemical probe, we have conducted activity-based proteomic profiling (ABPP) (Fig. 1c, d) and enriched chromosome 12 open reading frame 29 (C12orf29), a hitherto uncharacterised protein, from human cell lysates (Fig. 1e). C12orf29 is a 37 kDa human protein consisting of 325 amino acids. Sequence ana-lysis shows that it is highly conserved among the chordate superphylum[26] with C12orf29 orthologs being present in vertebrates. In contrast, we could not identify C12orf29 orthologs in any other eukaryotes, except for a few species including gastropods, bivalves, and cephalopods within the phylum Mollusca[26] (Supplementary Fig. 1). Our subsequent investigation identified C12orf29 as a human 5′–3′ RNA ligase that operates via a tandem auto- and RNA-AMPylation mechanism. Preliminary cellular studies also demonstrate its involve-ment in maintaining RNA integrity under reactive oxygen species (ROS)-induced cellular stress conditions. Overall, our work presents a chemistry-led functional identification of a previously unknown pro-tein as a human 5′–3′ RNA ligase, which suggests a latent RNA repair machinery.

## Results

### Discovery of C12orf29 by chemical proteomics dedicated to identifying AMPylated proteins

To study the human AMPylated proteome, we designed and synthe-sised a *C*2,*C*2′-ethynyl-modified Ap₃A probe (*C*2-eAp₃A) (Fig. 1b and see details in Supplementary Methods). Once incubated with whole-cell lysates, this chemical probe was expected to be used as a co-substrate in place of ATP during the AMPylation of cellular proteins. Conse-quently, a target protein would be covalently modified with a *C*2-alkynyl AMP group, which would allow the attachment of an affinity tag, des-thiobiotin (DB) (Fig. 1b-d), via Cu(I)-catalysed azide-alkyne cycloaddi-tion (CuAAC) for downstream ABPP. Among the proteins identified from both human non-small cell lung carcinoma cells (H1299) and human embryonic kidney cells (HEK293T), one caught our attention since it had not yet been characterised: C12orf29 (see details in Sup-plementary Fig. 2 and Supplementary Data 1, 2 and 3). Its enrichment from H1299 and HEK293T was further verified by immunoblotting of the elution fractions using an anti-C12orf29 antibody (Fig. 1e).

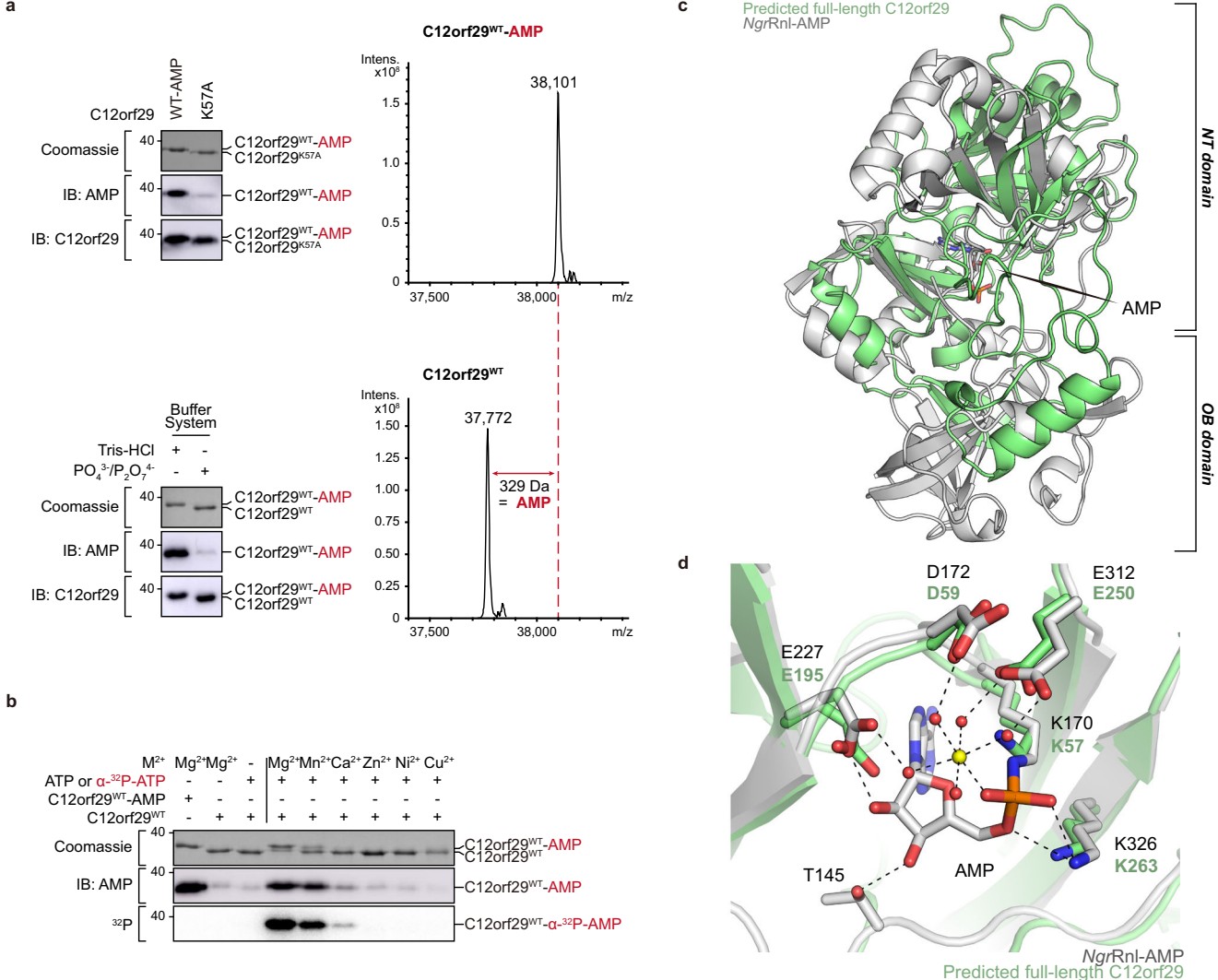

**Fig. 2 | Auto-AMPylation activity and structural prediction of C12orf29. a** (Top) immunoblotting of C12orf29[WT]-AMP and C12orf29[K57A] and LC-MS analysis of C12orf29[WT]-AMP. C12orf29[K57A] is deficient in auto-AMPylation activity (representative images of $n = 3$). Mass spectra indicating a 329 Da increase in mass upon auto-AMPylation of C12orf29[WT]. C12orf29[WT]-AMP: calc. 38,100 Da, found 38,101 Da. C12orf29[WT]: calc. 37,771 Da, found 37,772 Da. (Bottom) preparation of C12orf29[WT] and C12orf29[WT]-AMP with different buffer systems (representative images of $n = 3$). **b** Divalent metal ion dependency of auto-AMPylation activity of C12orf29[WT]. **c** Superimposition of the structure of C12orf29 predicted by AlphaFold (green)[31,51] on the structure of NgrRnl-AMP (PDB ID: 5COT, grey). OB, oligonucleotide-binding.

NT, nucleotidyltransferase. Structures were superposed in Coot[52] using structural equivalent residues identified by the DALI webserver[32]. **d** Enlarged view of the catalytic site of NgrRnl-AMP (PDB ID: 5COT, grey) and predicted C12orf29 (green). In the NgrRnl-AMP structure, AMP is covalently attached to the side chain of K170. D172, E227, and E312 bind $Mn^{2+}$ via water-mediated contacts and K326 contacts the phosphate moiety of AMP. Corresponding residues in the putatively catalytic site of C12orf29 are indicated. $Mn^{2+}$ and water molecules are depicted as yellow and red spheres, respectively. Atomic contacts are depicted as dashed lines. Source data are provided as a Source Data file.

To confirm the AMPylation of C12orf29, we prepared the recombinant wild-type (WT) C12orf29. Surprisingly, liquid chromatography-mass spectrometry (LC-MS) analysis revealed that the protein purified from *Escherichia coli* (*E. coli*) had a mass of 38,101 Da−329 Da higher than the calculated mass of 37,771 (found 37,772) Da for C12orf29[WT]−in consistence with an AMPylated form (Fig. 2a). Furthermore, the AMPylation of C12orf29 was validated by immunoblotting using a monoclonal antibody against AMPylation (Fig. 2a)[27]. Meanwhile, we also developed an expression and purification protocol that allowed the preparation of recombinant C12orf29 without AMPylation (Fig. 2a and see details in "Methods" section). The AMPylated and the non-modified form of C12orf29 showed different migration behaviours in an SDS-PAGE gel (Fig. 2a). With C12orf29[WT] and C12orf29[WT]-AMP in hand, we further proved that C12orf29 was auto-AMPylated using ATP as a co-substrate in the presence of $Mg^{2+}$ as the most proficient cofactor (Fig. 2b).

## C12orf29 is an RNA ligase operating via sequential auto- and RNA AMPylation

In parallel, we performed structure prediction of C12orf29 using bioinformatic tools. A putative structure of a truncated form of C12orf29 was obtained by Phyre2[28] based on the identity of 22 out of 71 residues (195-265) with those of *Naegleria gruberi* RNA ligase (NgrRnl)[29,30]. NgrRnl is a 5′−3′ RNA ligase and as such operates via the classic three-step mechanism (Fig. 1a)[29,30]. The structural similarity between C12orf29 and NgrRnl motivated us to investigate whether C12orf29 also possessed RNA ligase activity. Recently, AlphaFold[31] structure predictions for almost all proteins of the human genome were made available at https://alphafold.ebi.ac.uk[31]. We submitted the AlphaFold structure prediction for C12orf29, which was expected to be of high quality (average pLDDT = 91.4), to the Dali server[32], and compared it exhaustively to a representative subset of the Protein Data Bank. The results of structural superposition by Dali are highly

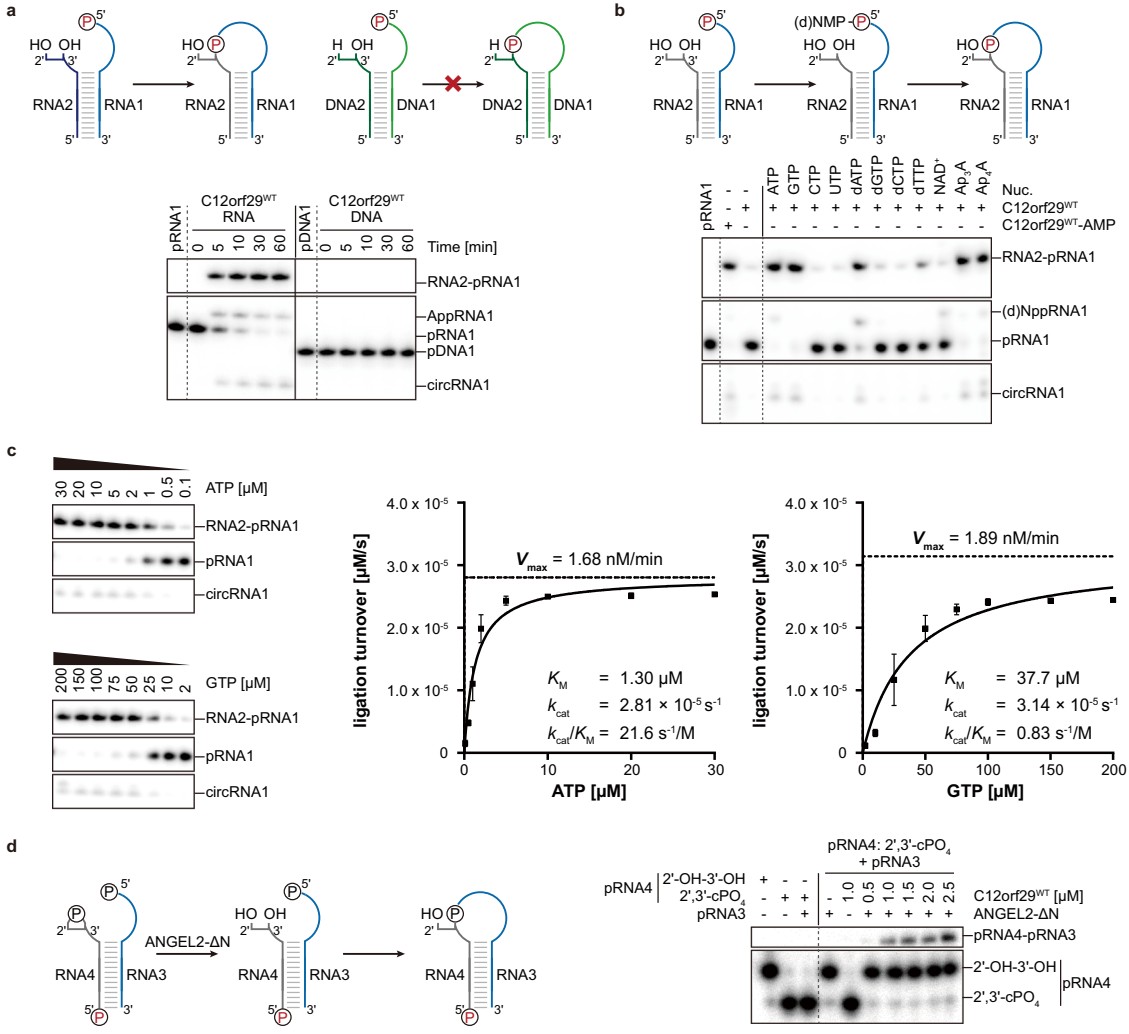

**Fig. 3 | RNA ligase activity of C12orf29.** RNA/DNA oligonucleotides are schematically depicted in blue/green, respectively. The $^{32}$P-labelled 5′-ends are depicted in red. The radioactive oligonucleotides were resolved by denaturing PAGE and analysed by phosphorimaging. **a** (Top) Schematic display of two nucleic acid substrates used. (Bottom) PAGE analysis of reaction products after incubation with C12orf29 for various time points as indicated. Ligation was observed for 5′-phosphorylated RNA (representative images of $n = 3$). **b** Reaction of C12orf29 with indicated RNA constructs using various nucleotides (representative images of $n = 3$). Besides ATP, C12orf29 is able to efficiently catalyse ligation by processing GTP, dATP, Ap$_3$A, and Ap$_4$A. Nuc., nucleotide. **c** (Left) C12orf29 promoted ligation reaction at various concentrations of ATP and GTP (representative images of $n = 3$). (Right) Michaelis-Menten fits to initial rates providing $K_M$, $k_{cat}$ and $k_{cat}/K_M$ for ATP and GTP. Plotted data represent the mean value ± SD for three biological replicates. **d** (Left) Proposed ligation scheme starting from 2′,3′-cyclic phosphorylated RNA by the sequential action of ANGEL2-ΔN and C12orf29$^{WT}$. (Right) PAGE analysis of the depicted reaction (representative images of $n = 3$). All oligonucleotide sequences are provided in Supplementary Table 1. Source data are provided as a Source Data file.

significant and display RNA ligases as the most similar structures (top hit, Z-score = 7.4 for *Ngr*Rnl), confirming our initial sequence-based suggestion for the functional assignment of C12orf29. In turn, the structure of C12orf29 predicted by AlphaFold was superimposed on *Ngr*Rnl (Fig. 2c, d and Supplementary Fig. 3). While an N-terminal oligonucleotide-binding (OB) domain as in *Ngr*Rnl is not observed for C12orf29, several secondary structure patterns in the predicted C12orf29 structure precisely overlap with the C-terminal nucleotidyltransferase (NT) domain of *Ngr*Rnl (Fig. 2c, d). Of note, C12orf29 contains a lysine at position 57 within the conserved sequence motif KX(D/H/N)G that defines the superfamily of nucleotidyltransferases (Supplementary Fig. 3)[33]. Indeed, mutation of the putatively critical lysine at position 57 to alanine (K57A) abolishes auto-AMPylation of the recombinant C12orf29 purified from *E. coli* (Fig. 2a).

Encouraged by the auto-AMPylation activity of C12orf29 in the presence of ATP and either Mg$^{2+}$ or Mn$^{2+}$ (Fig. 2b), we next investigated whether C12orf29 had 5′−3′ RNA ligase activity. We first assessed the nick-sealing activity of C12orf29 with a collection of double-stranded

substrates consisting of a DNA oligonucleotide splint annealed to two DNA or RNA oligonucleotides that were separated by a nick to be ligated. Although the nick-sealing activity was reported for *Ngr*Rnl[29], it was not observed with C12orf29 (Supplementary Fig. 4). In turn, we focused on single-stranded RNA (ssRNA) substrates and tested C12orf29 with constructs folded into an "open hairpin" (Fig. 3a). The 5′-terminus of one strand (17 nt) within the open loop was labelled with $^{32}$P-phosphate while the other strand (10 nt) was unlabelled (Fig. 3a and Supplementary Table 1). Successful ligation would lead to a 27-nt oligonucleotide with significantly altered migration within a denaturing PAGE gel. Indeed, we have found that C12orf29 is proficient in ligating the ssRNA overhangs to give a product that migrates at the expected length (Fig. 3a). Moreover, the formation of self-cyclised RNA as well as AppRNA was also observed. The enzyme exhibits profound selectivity for RNA:RNA constructs are efficiently ligated within short time while the respective DNA constructs are not detectably converted under the very same conditions (Fig. 3a). We also investigated whether nucleotides other than ATP represented proficient co-substrates and found

that other adenosine nucleotides such as diadenosine tri- and tetra-phosphate (Ap$_3$A, Ap$_4$A), and to a lesser extent even dATP were used as co-substrates, while NAD$^+$ was not used for promoting RNA ligation (Fig. 3b). Among the other nucleotides investigated, only GTP enables RNA ligation (Fig. 3b). Additional kinetic investigations, however, have revealed that the apparent catalytic efficiency ($k_{cat}/K_M$) of C12orf29 is about 29-fold higher with ATP than with GTP. This difference is mainly due to differences in $K_M$ while the $k_{cat}$ is similar for both nucleotides (Fig. 3c, Supplementary Fig. 5, and Supplementary Notes).

Next, we investigated if RNA substrates with a 3′-terminus bearing a 2′,3′-cyclic phosphate (cPO$_4$) or a 2′-phosphate end could be ligated by C12orf29. Neither construct was processed by the protein (Supplementary Fig. 6). However, when the RNA substrate with a 2′,3′-cPO$_4$ at the 3′-terminus was incubated in the presence of N-terminus-truncated ANGEL2 (ANGEL2-ΔN)—an enzyme recently reported to cleave 2′,3′-cPO$_4$ from the 3′-terminus to furnish a non-phosphorylated 3′-terminus[34]—we observed C12orf29-dependent ligation to occur (Fig. 3d). Altogether, these results show that non-phosphorylated 3′-RNA termini are essential for the C12orf29-mediated RNA ligation.

In order to investigate the structural requirements for C12orf29-catalysed RNA ligation within the single-stranded regions, we designed and studied single RNA oligonucleotides modified with $^{32}$P-phosphate at the 5′ ends, which folded into dumbbell structures bearing two single-stranded regions at the 5′ and 3′ ends (Fig. 4a, b). By varying the nucleotide composition at the ligation site, we have found that RNA constructs bearing purines at the ligation site are most efficiently processed (Fig. 4a). Moreover, RNA with longer 5′-overhangs are also ligated more efficiently (Fig. 4b).

Previous studies have identified several key residues at the catalytic centre of NgrRnl, which are responsible for substrate and cofactor coordination that underpinned the NgrRnl-mediated RNA ligation[29,30]. As shown in Fig. 2e and Supplementary Figs. 1 and 3, these residues are conserved in C12orf29 (i.e., K57, D59, E195, E250, K263), two of which are identified to be mutated in cancer cells (e.g., D59N[35], K263N[36]). In addition, mutations outside the catalytic site (e.g., R77L, E123D) are also found in patients suffering from glioblastoma[37] and chronic lymphocytic leukaemia[38], respectively. To examine the importance of these residues for RNA ligation, a series of C12orf29 variants bearing mutations at the respective positions were prepared and tested. Most of the point mutations are detrimental to RNA ligation (i.e., no detectable RNA ligation activity, Fig. 4c), whereas E123D and E123Q are fully active and R77L retained -12% activity.

## Knockout (KO) of C12orf29 impedes cellular resilience to reactive oxygen species

To gain insights into the function of C12orf29 in a cellular context, we generated human embryonic kidney (HEK293) cells, in which the *C12ORF29* gene (Supplementary Fig. 7) was knocked-out by CRISPR/Cas, and compared the properties of these cells with those of parental HEK293 cells expressing the WT enzyme. When cells were grown under physiological conditions, WT and KO cells showed similar phenotypes with respect to cell growth, adherence, and appearance (Fig. 5a). However, when cells were treated with up to 40 μM menadione—a molecule known to generate ROS-based cellular stress[39,40]—significant differences were observed in the viability levels of WT and KO cells (Fig. 5a). In comparison to WT cells, KO cells started to round up and detach at lower concentrations of menadione. Indeed, KO cells are more vulnerable to the treatment of menadione (Fig. 5b, c). For example, after a 3-hour treatment with 40 μM menadione, >85% WT cells were found to remain viable, whereas only <10% KO cells were alive. We also measured the level of ROS induced by 40 μM menadione over time by bioluminescence in both cell lines. Although a similar ROS level was found for KO cells after a 90-min treatment and WT cells after a 120-min treatment, the cell viability levels of the two cell lines differed by >50% (viability >85% for WT and <30% for KO) (Fig. 5c).

Next, we isolated and analysed the total RNA from the two cell lines treated with various concentrations of menadione (Fig. 6). Very prominent in this analysis are the signals of 28S and 18S rRNA. We found that the RNA degradation level depended on the concentration of the menadione applied. While the signal for 18S rRNA remained almost constant even at higher menadione concentrations, 28S rRNA appeared to be more susceptible to ROS damage. More importantly, in KO cells, 28S rRNA was found to start degrading upon treatment of 40 μM menadione. In contrast, 100 μM menadione treatment is required to induce significant 28S rRNA degradation in WT cells (Fig. 6 and Supplementary Table 2). Since the intracellular ROS level triggered by menadione treatment was similar in both cell lines (Fig. 5c), we attributed the more pronounced RNA decay in KO cells to the lack of C12orf29. This indicated that C12orf29 was important in maintaining RNA integrity under stress conditions.

## Discussion

Here, we have devised an Ap$_3$A-based chemical probe to interrogate the human AMPylated proteome. As a structural analogue of ATP, Ap$_3$A has shown superior chemical and enzymatic stability under reaction conditions[25], rendering Ap$_3$A-based probes promising alternatives to enrich AMPylated proteins from cell lysates without the need of external AMPylators[20–22]. The design of chemical probes based on dinucleoside polyphosphates can be potentially adopted by future research into protein NMPylation (e.g., UMPylation, GMPylation). Given the symmetrical design of the Ap$_3$A probe, it might be incorporated into cellular proteins upon both phosphorylation or AMPylation. We therefore note that care should be taken when interpreting the proteomic data to recognise potentially phosphorylated populations.

Using the Ap$_3$A-based chemical probe, we have discovered C12orf29 as a 5′–3′ RNA ligase in humans, which bears auto-AMPylation activity. We have shown in detail that C12orf29 ligates 5′-PO$_4$ and 3′-OH RNA termini within single-stranded regions via a three-step catalytic mechanism involving successive auto- and RNA-AMPylation. RNA lesions are generated by a transesterification reaction initiated by a 2′-OH and an adjacent 3′-5′-phosphodiester linkage to result in 2′,3′-cPO$_4$ and 5′-OH termini. So far, only a GTP-dependent 3′-5′ RNA ligase, RtcB (or HSPC117)[9], has been reported in human, which ligates such RNA termini via successive auto- and RNA-GMPylation[41,42]. Alternative multi-step repair pathways are seen across viruses, fungi, and plants, which involve a phosphatase and a kinase to convert 2′,3′-cPO$_4$ and 5′-OH termini to 3′-OH and 5′-PO$_4$ ends respectively for 5′−3′ RNA ligation[1,33,43]. Our discovery of a human 5′−3′ RNA ligase suggests a latent "healing and sealing" RNA repair mechanism in human.

Several nucleic acid ligases are known to be involved in repair of nucleic acids that are damaged e.g., by ROS[44]. Therefore, we investigated *C12ORF29*-KO cells in their response to ROS. In preliminary cellular studies, we have shown that *C12ORF29*-KO cells exhibit poor resilience to ROS with concurrent RNA (esp. rRNA) decay, which indicates the role of C12orf29 in maintaining RNA integrity as a cellular adaptive behaviour under oxidative stress conditions. rRNA accounts for >85% of the total RNA in most organisms as a core component of ribosomes[45], of which the synthesis is a major energy-consuming event in cells[46]. We reason that it is therefore energy- and time-saving for cells to rapidly restore functional rRNA by RNA repair, where C12orf29 may participate as an RNA ligase, rather than by de novo synthesis.

While C12orf29 does not appear to be necessary for cell viability under normal growth conditions, it is essential for maintaining RNA integrity and cell viability upon ROS-induced stress. In contrast, the RNA ligase RtcB and its homologues are reported to play vital roles in intron-containing tRNA maturation[9] and *X-box binding protein 1* (*XBP1*) mRNA splicing during UPR[3]. Furthermore, it has been suggested that the *E. coli* RtcB can repair nuclease- or antibiotic-induced rRNA damage[47,48]. The recently identified 2′,3′-cylic phosphatase ANGEL2

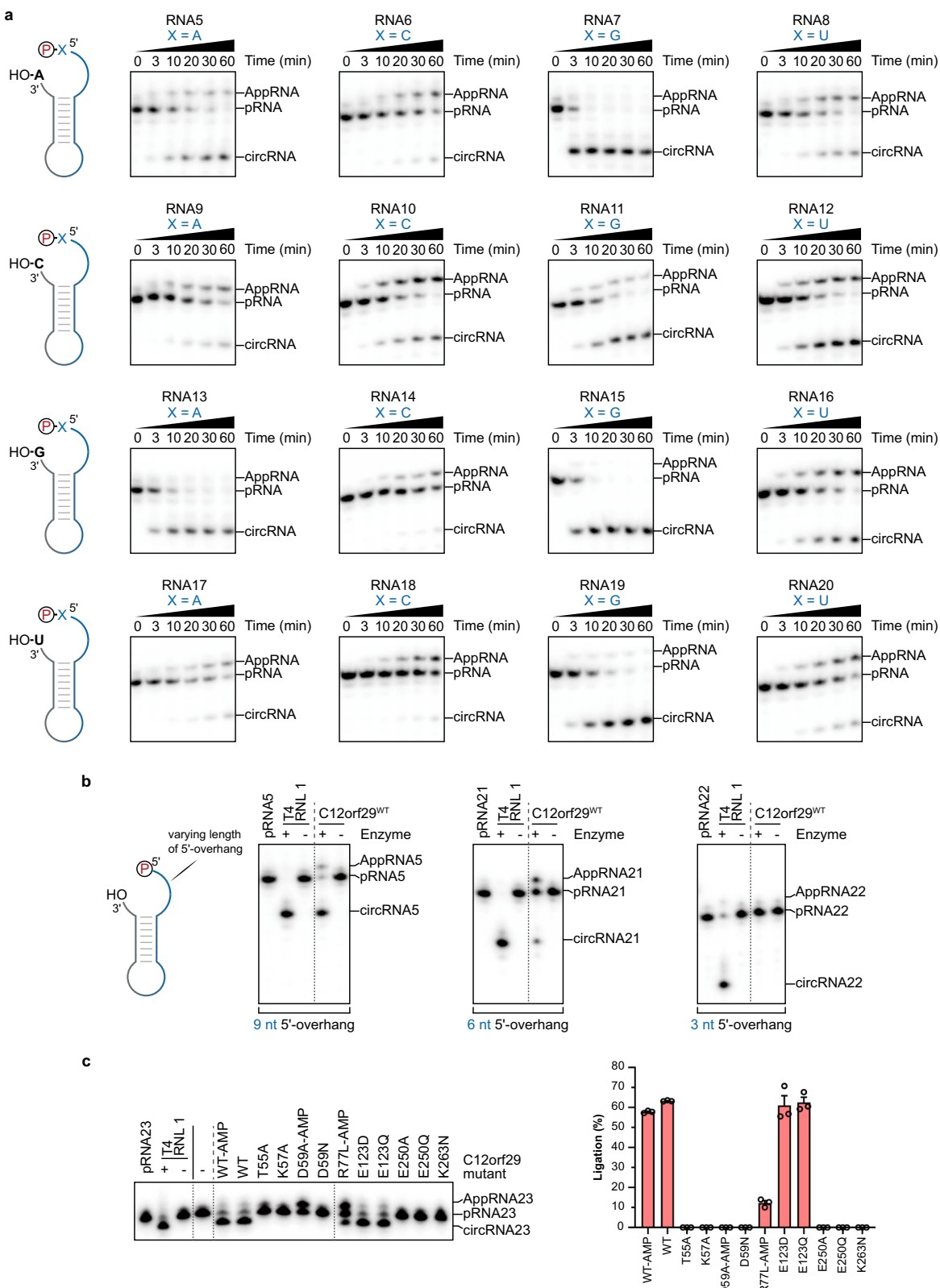

**Fig. 4 | Substrate scope of C12orf29.** RNA oligonucleotides that were used for intramolecular ligation resulting in cyclisation are schematically depicted. The [32]P-labelled 5′-ends were depicted in red. The radioactive oligonucleotides were resolved by denaturing PAGE and analysed by phosphorimaging. **a**, Investigation of the impact of the nucleobase composition at the 5′- and 3′- termini at the ligation site on the ligation efficiency of C12orf29 (representative images of $n = 3$). The depicted constructs were incubated under the same conditions for various time points. Most efficient ligation was observed when purines were at the ligation site. **b** Investigation of the impact of the length of the 5′-overhang on the ligation efficiency of C12orf29 (representative images of $n = 3$). **c** Impact of single site mutations on RNA ligase efficiency of C12orf29. Graph bars represent mean ligation efficiencies ± SEM and hollow circles represent individual data points for $n = 3$ biological replicates. All oligonucleotide sequences are provided in Supplementary Table 1. Source data are provided as a Source Data file.

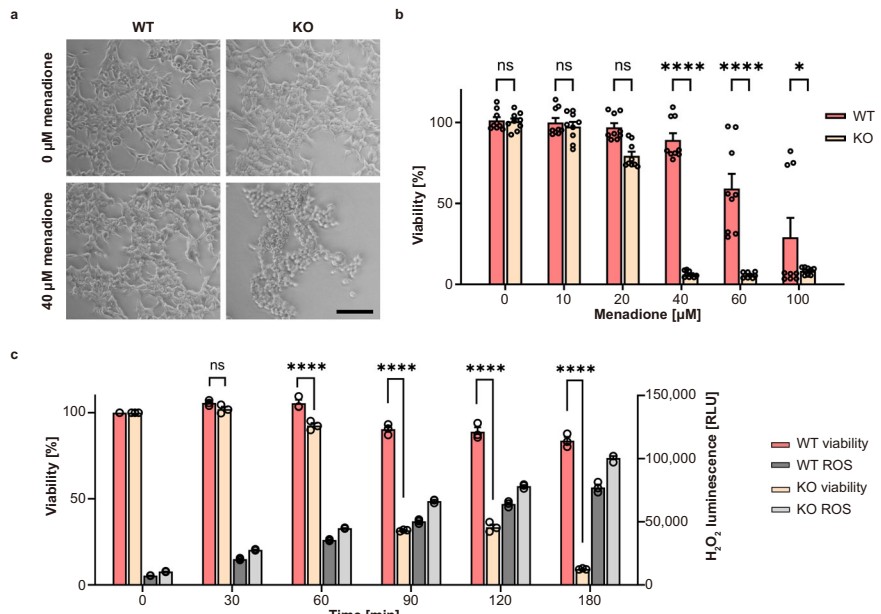

**Fig. 5 | Effect of menadione-induced oxidative stress on WT and C12orf29-KO cells. a** Light microscopy of HEK293 WT and *C12ORF29*-KO cells. The cells were treated with 40 μM menadione for 3 h or only with the carrier as control (0 μM menadione). Scale bars are 100 μm. **b** Cell viability of HEK293 WT and *C12ORF29*-KO cells after 3 h treatment with different menadione concentrations. Hollow circles represent individual data points for $n = 9$ biological replicates. Error bars represent the ± SEM. Significance was calculated by two-way ANOVA with Sidak's multiple comparisons test: $^{ns}P > 0.05$; $^{*}P ≤ 0.05$; $^{****}P ≤ 0.0001$. **c** Cell viability and corresponding $H_2O_2$ concentrations in HEK293 WT and *C12ORF29*-KO cells at different time points after treatment with 40 μM menadione. Hollow circles represent individual data points for $n = 3$ biological replicates. Error bars represent the ± SEM. Significance was calculated by two-way ANOVA with Sidak's multiple comparisons test: $^{ns}P > 0.05$; $^{****}P ≤ 0.0001$. RLU, relative light units. Source data are provided as a Source Data file.

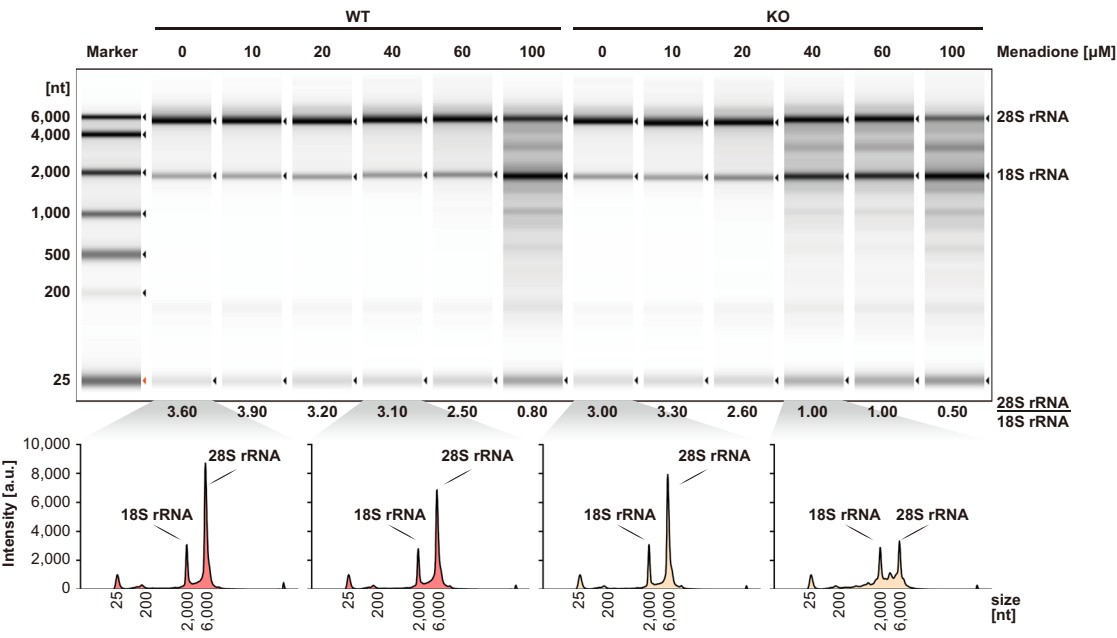

**Fig. 6 | Analysis of total RNA from cell extracts of HEK293 WT and *C12ORF29*-KO cells treated with various concentrations of menadione.** (Top) Cells (left: HEK293 WT cells; right: *C12ORF29*-KO HEK293 cells) were treated with different menadione concentrations for 3 h. In turn, total RNA was isolated and subsequently analysed by TapeStation (version 4.1.1). The ratio of 28S and 18S rRNA intensities were listed below the electrophoregram (a.u., arbitrary units). (Bottom) Electropherograms of the TapeStation analysis of total RNA from cell extracts of HEK293 WT (in red) and *C12ORF29*-KO (in yellow) cells. Source data are provided as a Source Data file.

can hydrolyse the 2′,3′-cPO$_4$ at 3′-termini to 2′-OH-3′-OH, thereby antagonising RtcB-mediated tRNA and *XBP1* mRNA splicing[34]. This antagonistic effect may suggest another layer of regulation of RNA processing events. Along with the 5′-OH RNA kinase *Hs*Clp1[49], RNA termini are processed to 2′-OH-3′-OH and 5′-PO$_4$, which can be recognised by C12orf29 for ligation. Therefore, we speculate that C12orf29 may be involved in an RNA ligation machinery in concert with other RNA processing enzymes when the RNA termini and activities are compromised in the RtcB-mediated RNA ligation system. Further work to decipher the endogenous interactors and RNA substrates of

C12orf29 are required to fully elucidate the C12orf29-mediated RNA ligation pathway.

Collectively, we have demonstrated a chemistry-led discovery of the human 5′–3′ RNA ligase C12orf29. Therefore, we propose to name the protein after *Homo sapiens* RNA ligase (*Hs*Rnl).

## Methods

### Cell culture and lysate preparation

HEK293T (ATCC) or H1299 (ATCC) cells were cultured in DMEM (Gibco™) supplemented with 10% (v/v) foetal bovine serum at 37 °C, 5% $CO_2$. Cells were harvested by centrifugation for 10 min at $500 \times g$ at 4 °C, and washed with ice-cold 1× PBS thrice. The supernatant was discard. Cell pellets were frozen in liquid nitrogen and stored at −80 °C.

Cell pellets were resuspended in lysis buffer (1× PBS pH 7.4, 1 mM EDTA, and 1× cOmplete™, EDTA-free protease inhibitor cocktail (Roche)) on ice and lysed by sonication. The lysates were cleared by centrifugation (30 min, $21{,}885 \times g$, 4 °C). The protein concentration of the supernatant was determined by bicinchoninic acid assay.

### Chemical proteomics towards the identification of C12orf29

The synthesis of the probes is detailed in the Supplementary Methods. The NMR data were processed by MestReNova (version 14.1.2-25024). In the chemical proteomics assay, 200 μM $Ap_3A$, $C2$-e$Ap_3A$, or MilliQ® $H_2O$ were incubated with 2.0 mg/mL H1299 or HEK293T cell lysates in 1× AMPylation buffer (20 mM HEPES pH 7.4, 100 mM NaCl, 5 mM $MgCl_2$, and 1 mM DTT) at 37 °C for 1 h in a total volume of 450 μL. The reaction was stopped by adding 1.8 mL pre-cold MeOH. The resulting mixture stood at −20 °C for 2 h to precipitate. Protein pellets were obtained after centrifugation at $14{,}000 \times g$ for 10 min at 4 °C, which were dried for 5 min and reconstituted in 450 μL 1× resuspension buffer (50 mM triethanolamine pH 7.4, 150 mM NaCl, and 4% SDS). A master mix was prepared freshly with 0.5 mM $N_3$-(Arg)$PEG_3$-DB, 2.5 mM $CuSO_4$, 0.25 mM TBTA, and 2.5 mM TCEP in 0.4× AMPylation buffer. Three hunderd microliters of the master mix was added to the pre-cold reaction mixture to yield 0.2 mM $N_3$-(Arg)$PEG_3$-DB, 1.0 mM $CuSO_4$, 0.1 mM TBTA, and 1.0 mM TCEP in a total volume of 750 μL. The CuAAC was conducted at 25 °C for 1 h, which was quenched by adding 3 mL pre-cold acetone. The resulting mixture stood at −20 °C overnight to precipitate. Protein pellets were obtained after centrifugation at $14{,}000 \times g$ for 10 min at 4 °C, which were washed with 300 μL cold MeOH trice and dried for 5 min. The pellets were reconstituted in 200 μL 1× PBS pH 7.4 supplemented with 4% SDS, followed by addition of 800 μL 1× PBS pH 7.4 and centrifugation at $12{,}000 \times g$ for 5 min at room temperature to remove any undissolved residue. The supernatant was incubated with high capacity streptavidin agarose beads in a bed volume of 25 μL at 25 °C for 15 min with an end-over-end rotator. The beads were pelleted by centrifugation at $150 \times g$ for 2 min at room temperature, which were washed successively with 1× PBS pH 7.4 supplemented with 1% SDS ($3 \times 100$ μL), washing buffer ($8 \times 100$ μL, 1× PBS pH 7.4, 150 mM NaCl, 4 M urea, and 1% SDS), and 50 mM $NH_4HCO_3$ pH 7.8 ($5 \times 100$ μL). The beads were treated with 0.8 mM biotin in 50 mM $NH_4HCO_3$ pH 7.8 supplemented with 0.1% RapiGest SF ($3 \times 50$ μL) and incubated at 37 °C for 10 min with shaking at 600 rpm to elute the AMPylated proteins. The elution fractions were kept on ice for downstream in-solution digestion (see below). Alternatively, the elution fractions were added 6x loading buffer and heated to 95 °C for denaturation and concentration. The resulting mixture was resolved by SDS-PAGE and subjected to immunoblotting. Mouse anti-C12orf29 antibody (Santa Cruz Biotechnology, sc-390730, 1:1000) and goat anti-mouse HRP-conjugated antibody (Jackson ImmunoResearch, 115-035-062, 1:30,000) were used as primary and secondary antibody, respectively.

Prior to MS analysis, the elution fractions were treated with 5 mM DTT at 60 °C for 1 h. After cooling down, the mixture was treated with 50 mM 2-chloroacetamide at room temperature for 1 h in dark. In turn, the resulting mixture was incubated with 3 μg trypsin at 37 °C overnight. The digested protein mixture was added 3 μL TFA and incubated at 37 °C for 45 min to hydrolyse RapiGest SF. The resulting mixture was cleared by centrifugation at $12{,}000 \times g$ for 30 min at room temperature. The supernatant was transferred carefully and lyophilised overnight. Finally, the tryptic peptides were dissolved in 60 μL 0.1% TFA supplemented with 10 μL 10% TFA and desalted using U-C18 ZipTips. Tryptic peptides were separated on an EASY-nLC 1200 system (Thermo Scientific) at a flow rate of 300 nl/min using a 39 min gradient from 2.5% MeCN/0.1% formic acid to 32% MeCN/0.1% formic acid, 1 min to 75% MeCN/0.1% formic acid, followed by a 5 min washing step at 75% MeCN/0.1% formic acid. Mass spectra were recorded on a QExactive HF mass spectrometer (Thermo Scientific) operated in data dependent Top20 mode with dynamic exclusion set to 5 s. Full scan MS spectra were acquired at a resolution of 120,000 (at $m/z$ 200) and scan range 350–1600 m/z with an automatic gain control target value of 3e6 and a maximum injection time of 60 ms. Most intense precursors with charge states of 2–6 reaching a minimum automatic gain control target value of 1e3 were selected for MS/MS experiments. Normalised collision energy was set to 28. MS/MS spectra were collected at a resolution of 30,000 (at $m/z$ 200), an automatic gain control target value of 1e5 and 100 ms maximum injection time. Each of the biological triplicates was measured as technical duplicates resulting in six measurements per condition ($Ap_3A$, $C2$-e$Ap_3A$ and bead control).

Raw files from LC–MS/MS measurements were analysed using MaxQuant (version 1.6.1.0) with match between runs and label-free quantification (LFQ) (minimum ratio count 1) enabled. The minimal peptide length was set to 5. For protein identification, the human reference proteome downloaded from the UniProt database (download date: 2018-02-22) and the integrated database of common contaminants were used. Data of different cell types were quantified in separate analyses. Further data processing was performed using Perseus software (version 1.6.15.0). Identified proteins were filtered for reverse hits, common contaminants and proteins that were only identified by site. LFQ intensities were log2 transformed, filtered to be detected in at least 4 out of 6 replicate measurements of at least one condition and missing values were imputed from a normal distribution (width 0.3 and shift 1.8), based on the assumption that these proteins were below the detection limit. LFQ intensities of technical replicates were averaged and significantly enriched proteins were identified by pairwise comparisons. For each cell type (HEK293T and H1299) two Student's $t$-tests ($S_0 = 0.1$ and FDR = 0.001) were performed, separately comparing $C2$-e$Ap_3A$ against $Ap_3A$ and against the bead control. Only proteins significantly enriched and showing a $t$-test difference >1 in both comparisons were considered to be $C2$-e$Ap_3A$-enriched for the cell type.

Chemical proteomics mass spectrometry data have been deposited to the ProteomeXchange Consortium via the PRIDE[50] partner repository with the dataset identifier PXD038132.

### Expression and purification of recombinant proteins

As for expression and purification of AMPylated C12orf29[WT] and its variants, plasmid constructs pET15b-C12orf29[WT] or plasmids containing inserts of C12orf29 variants were transformed in *E. coli* BL21 (DE3) competent cells, which were cultured in 50 mL LB medium containing 100 μg/mL carbenicillin at 37 °C, 180 rpm overnight. In turn, a defined volume of cell suspension was transferred to 1 L LB medium containing 100 μg/mL carbenicillin to reach $OD_{600} = 0.1$, followed by the incubation at 37 °C at 180 rpm until $OD_{600} = 0.7$. The mixture was cooled down on ice for 30 min and then incubated with 1.0 mM IPTG at 18 °C for 18 h at 180 rpm. Cells were harvested by centrifugation at $8000 \times g$ for 30 min at 4 °C. The pellet was resuspended in 30 mL cold lysis buffer (50 mM Tris-HCl pH 8.0, 150 mM NaCl, 1.0 mM DTT, 0.1% (v/v) Triton X-100, 20 mM imidazole, 1 μg/mL aprotinin, 1 μg/mL leupeptin,

and 1 mg/mL Pefabloc® SC) and lysed by sonication on ice. The lysates were centrifuged at 40,000 × $g$ for 30 min at 4 °C and filtered through 0.45 μm syringe filter. The N-terminal His$_6$-tagged AMPylated C12orf29 was purified using a 5 mL HisTrap™ FF crude column (Buffer A: 50 mM Tris-HCl pH 8.0, 150 mM NaCl, 1.0 mM DTT, and 20 mM imidazole; Buffer B: 50 mM Tris-HCl pH 8.0, 150 mM NaCl, 1.0 mM DTT, and 500 mM imidazole). Fractions containing AMPylated His$_6$-C12orf29 were pooled and dialysed against a buffer containing 50 mM Tris-HCl pH 8.0, 100 mM NaCl, 5 U/mg thrombin, and 1 mM DTT at 4 °C overnight. The resulting solution was further purified by anion IEX on a 5 mL HiTrap™ Q HP column (Buffer A: 50 mM Tris-HCl pH 8.0, 100 mM NaCl, and 1.0 mM DTT; Buffer B: 50 mM Tris-HCl, pH 8.0, 1000 mM NaCl, and 1.0 mM DTT). Pure fractions were pooled, concentrated, and stored at −20 °C in a storage buffer containing 25 mM Tris-HCl pH 8.0, 100 mM NaCl, 1 mM DTT, and 50% (v/v) glycerol.

As for expression and purification of deAMPylated C12orf29$^{WT}$ and its variants, plasmid constructs pET15b-C12orf29$^{WT}$ or plasmids containing inserts of C12orf29 variants were transformed in *E. coli* BL21 (DE3) competent cells, which were cultured in 50 mL LB medium containing 100 μg/mL carbenicillin at 37 °C, 180 rpm overnight. In turn, a defined volume of cell suspension was transferred to 1 L LB medium containing 100 μg/mL carbenicillin to reach OD$_{600}$ = 0.1, followed by the incubation at 37 °C at 180 rpm until OD$_{600}$ = 0.7. The mixture was cooled down on ice for 30 min and then incubated with 1.0 mM IPTG at 18 °C for 18 h at 180 rpm. Cells were harvested by centrifugation at 8000 × $g$ for 30 min at 4 °C. The pellet was resuspended in 30 mL cold lysis buffer (50 mM KH$_2$PO$_4$ pH 8.0, 10 mM Na$_4$P$_2$O$_7$, 150 mM NaCl, 1.0 mM DTT, 0.1% (v/v) Triton X-100, 20 mM imidazole, 1 μg/mL aprotinin, 1 μg/mL leupeptin, and 1 mg/mL Pefabloc® SC) and lysed by sonication on ice. The lysates were centrifuged at 40,000 × $g$ for 30 min at 4 °C and filtered through 0.45 μm syringe filter. The N-terminal His$_6$-tagged deAMPylated C12orf29 was purified using a 5 mL HisTrap™ FF crude column (Buffer A: 50 mM KH$_2$PO$_4$ pH 8.0, 10 mM Na$_4$P$_2$O$_7$, 150 mM NaCl, 1.0 mM DTT, and 20 mM imidazole; Buffer B: 50 mM KH$_2$PO$_4$ pH 8.0, 10 mM Na$_4$P$_2$O$_7$, 150 mM NaCl, 1.0 mM DTT, and 500 mM imidazole). Fractions containing His$_6$-C12orf29 were pooled and dialysed against a buffer containing 50 mM Tris-HCl pH 8.0, 100 mM NaCl, 5 U/mg thrombin, and 1 mM DTT at 4 °C overnight. The resulting solution was further purified by anion IEX on a 5 mL HiTrap™ Q HP column (Buffer A: 50 mM Tris-HCl pH 8.0, 100 mM NaCl, and 1.0 mM DTT; Buffer B: 50 mM Tris-HCl, pH 8.0, 1000 mM NaCl, and 1.0 mM DTT). Pure fractions were pooled, concentrated, and stored at −20 °C in a storage buffer containing 25 mM Tris-HCl pH 8.0, 100 mM NaCl, 1 mM DTT, and 50% (v/v) glycerol.

As for expression and purification of ANGEL2-ΔN, plasmid constructs pET15b-ANGEL2-ΔN were transformed in *E. coli* BL21 (DE3) cells, which were cultured in 50 mL LB medium containing 100 μg/mL carbenicillin at 37 °C, 180 rpm overnight. In turn, a defined volume of cell suspension was transferred to 1 L LB medium containing 100 μg/mL carbenicillin to reach OD$_{600}$ = 0.1, followed by the incubation at 37 °C at 180 rpm until OD$_{600}$ = 0.7. The expression was induced by the addition of 1.0 mM IPTG after cooling down the mixture on ice for 30 min. The mixture was then incubated at 18 °C for 18 h at 180 rpm before the cells were harvested by centrifugation at 8,000 × $g$ for 30 min at 4 °C. The pellet was resuspended in 30 mL cold lysis buffer (50 mM Tris-HCl pH 8.0, 100 mM KCl, 1 mg/mL lysozyme, 0.1% (v/v) Triton X-100, 1 mM DTT, 1 μg/mL aprotinin, 1 μg/mL leupeptin, and 1 mg/mL Pefabloc® SC) on ice for 45 min and sonicated. The lysate was centrifuged at 40,000 × $g$ for 30 min at 4 °C and filtered through 0.45 μm syringe filter. The His$_6$-tagged ANGEL2-ΔN was purified using a 5 mL HisTrap™ FF crude column (Buffer A: 50 mM Tris-HCl pH 8.0, 100 mM KCl, 1.0 mM DTT, and 20 mM imidazole; Buffer B: 50 mM Tris–HCl pH 8.0, 100 mM KCl, 1.0 mM DTT, and 500 mM imidazole). Fractions containing His$_6$-tagged ANGEL2-ΔN were pooled and dialysed against a buffer containing 50 mM Tris-HCl pH 8.0, 100 mM KCl,

5 U/mg thrombin, and 1 mM DTT at 4 °C overnight. The resulting solution was further purified by anion IEX on a 5 mL HiTrap™ Q HP column (Buffer A: 50 mM Tris-HCl pH 8.0, 100 mM KCl, and 1.0 mM DTT; Buffer B: 50 mM Tris-HCl, pH 8.0, 1000 mM KCl, and 1.0 mM DTT). Pure fractions were pooled, concentrated, and stored at −20 °C in a storage buffer containing 25 mM Tris-HCl pH 8.0, 50 mM KCl, 1 mM DTT, and 50% (v/v) glycerol.

### Identification of the intact protein mass
All samples were purified by UHPLC on a Dionex UltiMate3000 (Thermo Fisher Scientific, Germany) using an analytical Zorbax 300SB-C8 column (150 mm × 2.1 mm) with 3.5 μm silica as a stationary phase (Agilent, USA). Prior to purification, all samples were acidified with 10% TFA. Gradient elution (3 min at 0% Buffer B; in 19 min to 80% Buffer B; then in 8 min to 100% Buffer B) with Buffer A (0.02% TFA in water) and Buffer B (0.02% TFA in MeCN/water (80:20, v/v)) was performed at a flow rate of 300 μL/min. The signals were monitored by UV absorbance at 220 nm.

Intact proteins were then analysed by direct infusion on an amazon speed ETD mass spectrometer (Bruker Daltonics) with a flow rate of 4 μL/min. The mass spectrometric data were acquired for about 10 minutes and the final mass spectrum was averaged over the whole acquisition time. Mass spectrometric data were evaluated and deconvoluted using the Compass Data Analysis Version 4.4 (Bruker Daltonics) software.

### General procedure of C12orf29 auto-AMPylation assays
Unless otherwise noted, the C12orf29 auto-AMPylation assays were performed as follow. 1.0 μM C12orf29$^{WT}$-AMP, C12orf29$^{WT}$ or its variants were incubated with 200 μM ATP or a mixture of ATP:α-$^{32}$P-ATP (185 TBq/mmol, Hartmann Analytic, FP-307) in 9:1 ratio in 1× auto-AMPylation buffer (50 mM Tris-HCl pH 8.5, 5 mM MgCl$_2$, and 1 mM DTT) at 37 °C for 30 min in a total volume of 18 μL. The reaction was stopped by transferring 15 μL reaction mixture to a pre-cold PCR tube containing 0.43 μL 0.5 M EDTA, 0.36 μL 2 mg/mL BSA, and 3.16 μL 6x loading buffer (50 mM Tris–HCl pH 6.8, 10% (v/v) glycerol, 2% (w/v) SDS, and 1% (v/v) β-mercaptoethanol). The resulting mixture was heated at 95 °C for 5 min. Samples were resolved by SDS-PAGE and analysed by Coomassie staining, autoradiographic imaging, or immunoblotting. Mouse anti-AMPylation antibody[27] (1:1000) and goat anti-mouse HRP-conjugated antibody (Jackson ImmunoResearch, 115-035-062, 1:30,000) were used as primary and secondary antibody, respectively.

### Preparation of 5′ $^{32}$P-labelling of oligonucleotides
Oligonucleotides (1.0 μM) were incubated with 15 units of T4 PNK (New England BioLabs, M0202S) and 200 μM 0.555 MBq γ-$^{32}$P-ATP (185 TBq/mmol, Hartmann Analytic, SRP-401) in 1× T4 PNK reaction buffer at 37 °C for 1 h in a total volume of 15 μL. The reaction was stopped by heating to 95 °C for 2 min. The excess amount of γ-$^{32}$P-ATP was removed by gel filtration using Sephadex™ G-10 resin to give 5′ $^{32}$P-lablled oligonucleotides in a concentration of 1.0 μM. When labelling RNA oligos bearing 2′,3′-cPO$_4$ or 2′-PO$_4$-3′-OH on the 3′-ends, T4 PNK 3′ phosphatase minus (New England BioLabs, M0236S) was used.

### General procedures of RNA ligation assays
As for general procedure of RNA ligation with C12orf29, unless otherwise noted, the RNA ligation with C12orf29 were performed as follow. 0.1 μM 5′ $^{32}$P-labelled oligonucleotide substrates were incubated with 1 μM C12orf29$^{WT}$ or its variants and 200 μM ATP in 1× RNA ligation buffer (50 mM Tris-HOAc pH 7.0, 5 mM MgCl$_2$, and 1 mM DTT) at 37 °C 1 h in a total volume of 10 μL. The reaction was quenched by adding 10 μL stopping solution (80% (v/v) formamide, 20 mM EDTA, 0.025% (w/v) bromophenol blue, and 0.025% (w/v) xylene cyanol) and heating at 95 °C for 2 min. One microliter of the resulting mixture was further diluted to give 0.005 μM 5′ $^{32}$P-labelled oligonucleotides.

Samples were resolved by urea-PAGE and analysed by autoradiographic imaging.

In the time course study, the reaction was performed as stated above with C12orf29$^{WT}$-AMP and 1 U/µL recombinant RNasin® Ribonuclease Inhibitor (Promega) in a total volume of 20 µL. Aliquots (1.5 µL) were taken at 0, 3, 10, 20, 30, and 60 min and quenched by adding 28.5 µL stopping solution (50 mM EDTA pH 8.0). The resulting mixture was heated at 95 °C for 2 min. Samples were resolved by urea-PAGE and analysed by autoradiographic imaging.

### RNA ligation in the presence of ANGEL2-ΔN with RNA substrates bearing 2′,3′-cPO$_4$ on the 3′ ends

5.0 µM of the 5′-OH RNA oligo3 substrate was incubated with 10 units of T4 PNK and 1.0 mM ATP in 1x T4 PNK reaction buffer at 37 °C for 1 h in a total volume of 50 µl. The reaction was stopped by heating to 95 °C for 2 min. The excess of ATP was removed by gel filtration to yield the respective 5′ phosphorylated RNA oligo3 in a concentration of 5.0 µM. 0.5 µM of the $^{32}$P-labelled RNA oligo4 was mixed with 0.6 µM of the non-radioactively 5′-phosphorylated RNA oligo3 in MilliQ® H$_2$O. The RNA strands were annealed using the annealing programme described in general procedure.

0.1 µM of the annealed 5′ $^{32}$P-labelled RNA oligo4 and with 5′ non-radioactively phosphorylated RNA oligo3 complexes were incubated with different concentrations of C12orf29, 1 µM ANGEL2-ΔN, and 2.0 mM ATP in 1× RNA ligation buffer at 37 °C for 2 h in a total volume of 10 µL. The reaction was quenched by adding 95 µL stopping solution to 5 µL of the sample and heating to 95 °C for 2 min. The samples were then resolved by urea-PAGE (12%) and analysed by autoradiographic imaging.

### Light-microscopy of menadione-treated HEK293 cells

$1.2 \times 10^6$ cells were seeded in 4 mL DMEM GlutaMAX$^{TM}$ medium (Gibco$^{TM}$, Thermo Fisher) supplemented with 10% (v/v) FCS on 6 cm cell culture dishes (Sarstedt). After 48 h, the cells were treated directly with either 4 µL EtOH (control, no menadione) or with 4 µL of 40 mM menadione in EtOH (final concentration = 40 µM menadione). After 3 h, pictures were taken with a light microscope using a 5× objective.

### Cell viability assay of HEK293 cells treated with different concentrations of menadione

The CellTiter-Glo® Luminescent Cell Viability Assay (Promega) was used according to the manufacturer's instruction. $4.0 \times 10^4$ cells per well were seeded one day before the menadione treatment in 90 µL DMEM GlutaMAX$^{TM}$ medium with 10% (v/v) FCS in a 96-well plate (Sarstedt). The plate was incubated at 37 °C, 5% CO$_2$, and 100% humidity for 24 h.

1000× menadione stock solutions were prepared in ethanol and frozen in aliquots. On the day of the treatment, aliquots were thawed and diluted 1/100 in MilliQ® H$_2$O. The cells were treated with 10 µL of different concentrations of menadione in MilliQ® H$_2$O. The plate was incubated at 37 °C, 5% CO$_2$, and 100% humidity. After 3 h, the cells were equilibrated to RT (15 min) and 100 µL of CellTiter-Glo® reagent was added per well and mixed thoroughly. The plate was incubated for 10 min on a shaker. 100 µL of the solution was transferred to a black 96-well plate and luminescence read-out was performed with a plate reader (PerkinElmer Victor3$^{TM}$ Multilabel Counter 1420).

The luminescence values of the treated cells were compared to the value of the cells of the control treatment, which were treated with 0 µM menadione (equals 100% viability), to give the calculated cell viability.

### RNA integrity analysis of HEK293 WT and KO cells treated with different concentrations of menadione

$5.0 \times 10^5$ cells were seeded in 2 mL DMEM GlutaMAX$^{TM}$ medium with 10% (v/v) FCS in a 6-well plate (Sarstedt) 48 h prior to the experiment.

1000× menadione stock solutions were prepared in ethanol and frozen in aliquots. Two microliters of the respective menadione stock was given to the cells resulting in the final desired menadione concentration for the treatment. After 3 h the cells were scraped down in the present medium and centrifuged (500 × g, 5 min, 4 °C). The cell pellet was washed with 1 mL ice-cold PBS and centrifuged again (500 × g, 5 min, 4 °C). The RNA was then extracted from the pellet using Quick-RNA Miniprep Kit (Zymo Research) with the provided in-column DNase I digest. The resulting RNA was analysed using an Agilent 4150 TapeStation system.

### Cell viability assay in combination with ROS assay

The ROS-Glo$^{TM}$ H$_2$O$_2$ Assay (Promega) was used according to the manufacturer's instruction. $4.0 \times 10^4$ cells per well were seeded one day before the menadione treatment in 70 µL DMEM GlutaMAX$^{TM}$ medium with 10% (v/v) FCS in a 96-well plate (Sarstedt). The plate was incubated at 37 °C, 5% CO$_2$, and 100% humidity for 24 h. 3 h before the final readout, 20 µL of the provided H$_2$O$_2$ substrate in H$_2$O$_2$ substrate dilution buffer was directly added to the cells in their growing medium (final concentration was 25 µM). Afterwards, 10 µL of menadione stock solution in EtOH was added to give the desired end concentration in a final volume of 100 µL. The cells were incubated for the desired stress time at 37 °C. The cells were equilibrated at RT for 15 min.

(A) For detection of ROS, 50 µL of the present medium of each well was transferred in a new, black 96-well plate and 50 µL of the provided ROS detection solution was added. After 20 min incubation at RT, the luminescence was measured using a plate reader (PerkinElmer Victor3$^{TM}$ Multilabel Counter 1420).

(B) For the cell viability assay, 50 µL of CellTiter-Glo® reagent was added to the remaining 50 µL medium in the 96-well plate and mixed thoroughly. The plate was incubated for 10 min on a shaker. 100 µL of the solution was transferred to a black 96-well plate and the luminescence was measured using a plate reader (PerkinElmer Victor3$^{TM}$ Multilabel Counter 1420).

## Data availability

The data that support this study are available from the corresponding authors upon request. The study made use of the following publicly available data set: PDB entry: [https://doi.org/10.2210/pdb5COT/pdb] The mass spectrometry proteomics data generated in this study have been deposited in the ProteomeXchange Consortium via the PRIDE partner repository[50] under accession code PXD038132. Figures with associated raw data: Figs. 3c, 4c, 5b, c, 6 and Supplementary Fig. 5. Specific data P values are also included within the Source Data file. Source data are provided with this paper.

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

## Acknowledgements

We thank A. Marquardt and A. Sladewska-Marquardt (Proteomics Center, University of Konstanz) for assistance in mass spectrometry. Y.Y., F.M.S., L.A.S., P.S., L.B.H., M.F., and E.H. thank the Konstanz Research School Chemical Biology for support. We thank D. Höpfner and A. Itzen, UKE Hamburg, for the provision of the anti-AMPylation antibody. A.M. acknowledges the European Research Council for funding of this project (ERC AdG AMP-Alarm, 101019280). F.S. acknowledges the Deutsche Forschungsgemeinschaft for funding (STE 2517/5-1). O.P.S. acknowledges support by the Alexander von Humboldt Foundation.

## Author contributions

Y.Y., F.M.S., L.A.S., O.P.S. and A.M. conceived the study and experimental approach; Y.Y., F.M.S., L.A.S., O.P.S. generated C12orf29 and its mutants, performed AMPylation and RNA ligation experiments, Y.Y. and P.S. conducted the modelling and MS analysis. F.M.S. conducted cellular experiments. L.B.H. and M.F. provided expert expertise in PAGE analysis and AMPylation reactions, respectively. M.S., E.H., F.S., K.D. provided expert expertise on cellular studies, mass spectrometry and modelling, respectively. Y.Y., F.M.S., L.A.S., O.P.S. and A.M. analysed the data, A.M. wrote the manuscript with input from all authors.

## Funding

## Competing interests

The authors declare no competing interests.
