## [Peer Review File · Nature Communications]

REVIEWER COMMENTS

Reviewer #1 (Remarks to the Author):

The manuscript entitled "Chemoproteomic discovery of a human RNA ligase" describes previously uncharacterized protein C12orf29, which is able to perform RNA ligation reactions via a classical 3-step mechanism. The presented data identifies C12orf29 as the first human 5'-3' RNA ligase. This discovery is very interesting, and this reviewer agrees about the importance of this finding. However, several points should be addressed for publication in Nature Communications that further clarifications are required.

Detailed comments:

1. Proteomics results should be carefully examined by paired comparison. The list of AMPylated proteins found in the ABPP experiment should be included in a separate table. The authors should compare their protein list with already known AMPylated proteins as well as give a general description of their findings in this experiment.
2. It would be better to add rationale underneath about the selection of C12orf29 from the AMPylated protein list for follow-up study from the chemical proteomic profiling.
3. There is a lack of logical flow in part of the text that corresponds to Fig. 2. Firstly, the authors explain the 3-step catalytic mechanism of 5'-3' RNA ligases in the introduction. The mechanism can be greatly illustrated by Fig. 2d. This reviewer suggests separating Fig. 2d and moving it to the introduction. Secondly, in the second paragraph of the Results part authors refer readers to Fig. 2b, which includes the result of immunoblotting of C12orf29-WT and C12orf29-K57A mutant. However, the authors explained how they found the key residue K57 later in the text, so the logical flow in this part of the text is not maintained. The authors should carefully re-write this part of the manuscript.
4. It should carefully present the significant figures and resolution in the mass value, particularly for the intact protein mass.
5. The authors didn't test all possible nucleobase pairs at the ligation site, Fig. 4a. Thus, they can't conclude that purines are ligated with the highest efficiency. Moreover, not only terminal nucleotides can impact ligation efficiency. The authors should check RNAs with different combinations of 1st, 2nd, and 3rd nucleotides at the 5' and 3' end of RNA. (this can be a difficult and time-consuming experiment)
6. The sequence data of mutated C12orf29 should be included in supporting information. It is highly recommended to provide a full sequence map of the plasmids. If the authors deposit the plasmid, that information also should be clearly stated.
7. There is no explanation why the authors chose menadion which is known to generate ROS-based cellular stress for their cell experiments. It seems necessary to explain better why the authors thought that C12orf29 might be related to cellular stress.
8. The statistically significant difference is not shown in Fig. 5c. All statistically significant differences should be marked.
9. The gel in Fig. 6 should be improved for publishable quality of the 28S RNA decay rate differences in WT and KO HEK293 cells. The table in Supplementary Fig. 8 looks better representation of this finding.

Reviewer #2 (Remarks to the Author):

In this paper, the Marx lab fills a longstanding knowledge gap in vertebrate RNA metabolism by identifying a human RNA ligase enzyme C12orf29 capable of joining 3'-OH and 5'-PO₄ termini. C12orf29 adheres to the canonical pathway employed by T4 RNA ligases that entails formation of ligase-AMP and AppRNA intermediates. Thus, humans and other vertebrates have two flavors of RNA ligase: the classic-type (C12orf29) and the RtcB-type, which joins 3'-PO₄ and 5'-OH ends via an entirely different chemical mechanism.

The work here is noteworthy for the clever route of discovery. Rather than looking for and purifying the enzyme based on ligase activity, they identified C12orf29 as an AMPylated species via its reaction with an alkyne-modified substrate c2-eAp3A. Excellent use is made of biophysical and biochemical methods to characterize the ligase and the ligase-AMP

adduct. These experiments nicely interrogated the RNA and nucleotide substrate specificities of the human ligase.

Going the extra mile, the authors conduct an initial characterization of a HEK293 C12orf29-knockout cell line and report that ablation of the ligase confers sensitivity to killing and rRNA damage by ROS-inducer menadione, suggestive of a role for C12orf29 in the repair of stress-associated RNA damage.

This is an important study that advances the field of RNA repair. It will be of great interest to the RNA community.

There are several issues, scientific and textual, that require attention, as listed below.

Comments:

1) The authors should provide more information regarding the proteins identified as AMPylated after modification by c2-eAp3A, affinity isolation, and MS analysis. How many AMPylated proteins were identified? Which ones are most abundant? How abundant was C12orf29 versus others. A list of the top ten, or more, would be valuable, to provide some perspective. Perhaps, a volcano plot of peptide enrichment in the c2-eAp3A pool versus the Ap3A pool.

2) The "hypothesis" on p. 5 regarding RNA transesterification doesn't really make sense and should be deleted. It is obvious that if the ligase does not work on nicked double-strand nucleic acid, then one would immediately proceed to test ssRNA, without need for any hypothesis, given that many classic RNA ligases (T4 Rnl1, yeast Trl1) are dedicated to sealing ssRNA termini. Delete text from "In our further . . . beneficial to a cell." The edited paragraph will flow smoothly.

3) Is it possible that the 29-fold higher apparent K_m for GTP versus ATP reflects low level of contamination of the commercial GTP with ATP? Is there direct evidence for formation of a ligase-GMP intermediate?

4) It would be valuable to be more expansive in the Discussion on the coexistence of RtcB and C12orf29 type ligase in human/vertebrate cells. It is important to note that RtcB is essential for cell viability, because it is the agent of tRNA splicing, and is also essential for mRNA splicing in the unfolded protein response. By contrast, the present study makes clear that C12orf29 is not required for cell viability or (one surmises) for any essential RNA transaction when cells are grown in culture under standard conditions.

5) I think the case for C12orf29 as a therapeutic target is weak and the text on this point ought to be deleted from the Abstract and the Discussion. The paper is interesting enough without such embellishment.

6) Introduction, lines 1-3. "mRNA alternative splicing" does not involve an RNA ligase. (The spliceosome catalyzes transesterification and is usually not considered to be a ligase, in that it does not synthesize a phosphodiester bond.) "rRNA maturation" does not involve an RNA ligase in most taxa. What is the evidence that an RNA ligase seals irradiation-induced RNA damage? These sentences (and citations) need reconsideration.

7) p. 3. The authors briefly discuss the phylogenetic distribution of C12orf29 in the Introduction, with cursory reference to presence in "higher eukaryotes like vertebrates" and absence in "lower eukaryotes such as yeast". The high/lower distinction is vague here as other arguably "higher" eukarya (i.e., metazoa) lack this protein. The phylogeny deserves more specific mention, either in Intro or in Discussion. The key points are that orthologs of C12orf29 protein are found in mammals, birds, reptiles, amphibians, and fishes (vertebrates) . . . and in a few invertebrates (mollusks). It is not present in not found in insects, arachnids, crustaceans, corals, worms, jellyfishes, sponges (invertebrate metazoa) or fungi.

Text edits:

a) Do not start sentences with subjective opinion adverbs (unrelated to sentence verb). -delete "Interestingly" (p. 4 and p. 6); "Intriguingly" (p5)

- b) p. 3, line 16: delete "including humans" (redundant, as included in "higher eukaryotes")
- c) p. 3, lines 33-34: should be "highly conserved among vertebrates but absent in yeast" (delete "higher eukaryotes like" and "lower eukaryotes species such as")
- d) p. 3, line 28: delete "increasingly"
- e) p. 5, line 17: should be "patterns . . . overlap" (not overlaps)

Reviewer #3 (Remarks to the Author):

The manuscript by Yuan et al reports an important novel enzyme function of the previously uncharacterized protein C12orf29 as the first known RNA ligase in human cells. This groundbreaking discovery is based on a chemical proteomic profiling with a C2-Ap3A probe (dimeric ATP with alkyne modification) which was expected to be used as a surrogate for ATP and transferred onto proteins. C12orf29 appeared as hit and was first confirmed to be modified with AMP. Further in-depth studies revealed a structural homology (based on alpha fold) to a RNA ligase (NgrRnl) and corresponding RNA ligation assays confirmed this notion. Moreover, in order to decipher the cellular role of this RNA ligase the authors made a knockout strain which was significantly more susceptible to oxidative stress resulting in enhanced degradation of 28S RNA further validating its crucial role for RNA integrity. The paper is well written and contains reports an exciting new enzyme functionality. The following points require some attention:

1. The probe design and proteomic results need improved clarity. The authors mention in the discussion that their probes do not require external AMPylators for transfer and are thus superior to other methods. Recent methods with cell-permeable AMP prodrugs should be mentioned here as well as they utilize the cellular machinery in situ and also overcome these limitations. In addition, the manuscript does not report any proteomic results and does not provide a rationale for the selection of C12orf29. Corresponding MS data, e.g. volcano plots of the enrichment should be shown at least in the SI and the raw data deposited at a public repository. What other proteins are identified with the C2-Ap3A probe and are any known AMPylators among the hits? This information would be important to better understand the selection of C12orf29 as well as the overall performance of the probe.
2. The authors performed kinetic experiments with C12orf29 and determine k_{cat} and K_M values. It would be interesting to compare these values with other RNA ligases. Is the performance in a similar range and if it is different, it would be interesting to discuss possible reasons?

POINT-BY-POINT RESPONSE TO THE REVIEWER COMMENTS

Reviewer #1 (Remarks to the Author):

The manuscript entitled “Chemoproteomic discovery of a human RNA ligase” describes previously uncharacterized protein C12orf29, which is able to perform RNA ligation reactions via a classical 3-step mechanism. The presented data identifies C12orf29 as the first human 5’-3’ RNA ligase. This discovery is very interesting, and this reviewer agrees about the importance of this finding.

We are very pleased with this overall positive evaluation.

However, several points should be addressed for publication in Nature Communications that further clarifications are required.

In the revision we have carefully addressed all points. This is described below.

Detailed comments:

1. Proteomics results should be carefully examined by paired comparison. The list of AMPylated proteins found in the ABPP experiment should be included in a separate table. The authors should compare their protein list with already known AMPylated proteins as well as give a general description of their findings in this experiment.

This assessment is equal to those of the other two reviewers. We apologize for not having included the proteomics data as part of the originally submitted manuscript. We now list all significantly enriched interactors in a separate SI-table (as Excel file, Supplementary Data 4) and have added paired comparisons in the Supporting Information (Supplementary Fig. 2). We have also deposited the raw files of the complete experimental dataset on the open access platform PRIDE. However, we decided against a lengthy description of our proteomics study in the main part of the manuscript, as we do not want to shift the focus from what we consider to be the most exciting part of this work, i.e. the discovery of a new RNA ligase in human cells.

2. It would be better to add rationale underneath about the selection of C12orf29 from the AMPylated protein list for follow-up study from the chemical proteomic profiling.

C12orf29 was one of the completely uncharacterized proteins we identified in the proteomics data. This sparked our interest in deciphering its function. To reflect the motivation, we write in the manuscript: “Among the proteins identified from both human non-small cell lung carcinoma cells (H1299) and human embryonic kidney cells (HEK293T),

one caught our attention since it had not yet been characterised: C12orf29 (see details in Supplementary Information and Supplementary Fig. 2).”

3. There is a lack of logical flow in part of the text that corresponds to Fig. 2. Firstly, the authors explain the 3-step catalytic mechanism of 5’-3’ RNA ligases in the introduction. The mechanism can be greatly illustrated by Fig. 2d. This reviewer suggests separating Fig. 2d and moving it to the introduction. Secondly, in the second paragraph of the Results part authors refer readers to Fig. 2b, which includes the result of immunoblotting of C12orf29-WT and C12orf29-K57A mutant. However, the authors explained how they found the key residue K57 later in the text, so the logical flow in this part of the text is not maintained. The authors should carefully re-write this part of the manuscript.

We agree with the reviewer and are grateful for this comment. To better reflect the logical flow in the order of the figures, we have fundamentally changed the arrangement in Figures 1 and 2 (e.g., moved 3-step catalytic mechanism of 5’-3’ RNA ligases to the introduction and Fig. 1A) as well as the corresponding text. We hope that the changes are in accordance with the reviewer's expectations.

4. It should carefully present the significant figures and resolution in the mass value, particularly for the intact protein mass.

We agree with the reviewer. The significant figures in the mass values have been adjusted in Fig. 2a and in the text of the manuscript.

5. The authors didn’t test all possible nucleobase pairs at the ligation site, Fig. 4a. Thus, they can’t conclude that purines are ligated with the highest efficiency.

We agree with the reviewer's assessment. To address this, we have now added new experiments in which we examined all combinations at the first positions. The $4^2 = 16$ combinations are included in the revised Fig. 4a (sequences were added to the SI). The results of these experiments are fully consistent with those originally included in the manuscript and now confirm that purines at the ligation sites do indeed promote the enzyme reaction.

Moreover, not only terminal nucleotides can impact ligation efficiency. The authors should check RNAs with different combinations of 1st, 2nd, and 3rd nucleotides at the 5’ and 3’ end of RNA.(this can be a difficult and time-consuming experiment)

We agree with the reviewer that this would be an interesting future experiment. However, we believe that investigating the further, second and third positions in the RNA at the ligation site is clearly beyond the scope of this manuscript as the effort and cost are disproportionate to the expected gain in knowledge. In fact, $4^4 = 256$ oligonucleotides would have to be examined for the two (5’, 3’) 1st and 2nd sites, and $4^6 = 4096$ oligonucleotides for

the two 1st to 3rd sites. The reviewer also acknowledged that such an endeavor will be difficult and time consuming (on top very costly) and, thus, we hope that the reviewer agrees with our view.

6. The sequence data of mutated C12orf29 should be included in supporting information. It is highly recommended to provide a full sequence map of the plasmids. If the authors deposit the plasmid, that information also should be clearly stated.

The sequence data of the C12orf29^{WT}, its mutants, ANGEL2-ΔN and AtRNL-CPD are now included in the Supporting Information. The plasmid maps of C12orf29^{WT}, ANGEL2-ΔN, and AtRNL-CPD are included in the Supporting Figures.

7. There is no explanation why the authors chose menadion which is known to generate ROS-based cellular stress for their cell experiments.

Menadione is known to cause ROS-based stress and is used in experiments investigating it (see references cited in the manuscript). Therefore, it seemed to us to be a good option - in any case, we were not aware of anything that would speak against it. Since the concentration of menadione can be adjusted very easily and the compound is more stable in stock solutions (compared to H₂O₂), it further seemed to us to be a suitable option to generate ROS.

It seems necessary to explain better why the authors thought that C12orf29 might be related to cellular stress.

To address this, we have added the following in the Discussion: "Several nucleic acid ligases are involved in repair of nucleic acids damaged e.g., by ROS⁴³. Therefore, we investigated C12ORF29-KO cells in their response to ROS."

8. The statistically significant difference is not shown in Fig. 5c. All statistically significant differences should be marked.

We agree with the reviewer's assessment. We have marked the significant differences in the Figure.

9. The gel in Fig. 6 should be improved for publishable quality of the 28S RNA decay rate differences in WT and KO HEK293 cells. The table in Supplementary Fig. 8 looks better representation of this finding.

We have amended the quality of the figure and hope that it is now suitable for publication.

Reviewer #2 (Remarks to the Author):

In this paper, the Marx lab fills a longstanding knowledge gap in vertebrate RNA metabolism by identifying a human RNA ligase enzyme C12orf29 capable of joining 3'-OH and 5'-PO₄ termini. C12orf29 adheres to the canonical pathway employed by T4 RNA ligases that entails formation of ligase-AMP and AppRNA intermediates. Thus, humans and other vertebrates have two flavors of RNA ligase: the classic-type (C12orf29) and the RtcB-type, which joins 3'-PO₄ and 5'-OH ends via an entirely different chemical mechanism. The work here is noteworthy for the clever route of discovery. Rather than looking for and purifying the enzyme based on ligase activity, they identified C12orf29 as an AMPylated species via its reaction with an alkyne-modified substrate c2-eAp3A. Excellent use is made of biophysical and biochemical methods to characterize the ligase and the ligase-AMP adduct. These experiments nicely interrogated the RNA and nucleotide substrate specificities of the human ligase.

Going the extra mile, the authors conduct an initial characterization of a HEK293 C12orf29-knockout cell line and report that ablation of the ligase confers sensitivity to killing and rRNA damage by ROS-inducer menadione, suggestive of a role for C12orf29 in the repair of stress-associated RNA damage.

This is an important study that advances the field of RNA repair. It will be of great interest to the RNA community.

We are very pleased with this overall positive evaluation.

There are several issues, scientific and textual, that require attention, as listed below.

In the revision we have carefully addressed all points. This is described below.

Comments:

1) The authors should provide more information regarding the proteins identified as AMPylated after modification by c2-eAp3A, affinity isolation, and MS analysis. How many AMPylated proteins were identified? Which ones are most abundant? How abundant was C12orf29 versus others. A list of the top ten, or more, would be valuable, to provide some perspective. Perhaps, a volcano plot of peptide enrichment in the c2-eAp3A pool versus the Ap3A pool.

As written already above in response to reviewer 1, this assessment is equal to those of the other two reviewers.

We apologize for not having included the proteomics data as part of the originally submitted manuscript. We now list all significantly enriched interactors in a separate SI-table (as Excel file, Supplementary Data 4) and have added paired comparisons in the Supporting Information (Supplementary Fig. 2). We have also deposited the raw files of the complete experimental dataset on the open access platform PRIDE. However, we decided against a

lengthy description of our proteomics study in the main part of the manuscript, as we do not want to shift the focus from what we consider to be the most exciting part of this work, i.e. the discovery of a new RNA ligase in human cells.

2) The “hypothesis” on p. 5 regarding RNA transesterification doesn’t really make sense and should be deleted. It is obvious that if the ligase does not work on nicked double-strand nucleic acid, then one would immediately proceed to test ssRNA, without need for any hypothesis, given that many classic RNA ligases (T4 Rnl1, yeast Trl1) are dedicated to sealing ssRNA termini. Delete text from “In our further . . . beneficial to a cell.” The edited paragraph will flow smoothly.

We agree with the reviewer and changed the text as suggested.

3) Is it possible that the 29-fold higher apparent Km for GTP versus ATP reflects low level of contamination of the commercial GTP with ATP? Is there direct evidence for formation of a ligase–GMP intermediate?

This is a very good point. In order to investigate this, we have carefully analyzed the quality of the GTP stock (purchased from Jena Bioscience) and found that the GTP stock contained approx. 0.2% ATP. GTP was used at 200uM, which would result in an ATP concentration of 0.4uM. At this concentration, the ligase is not significantly active e.g., as can be seen in Figure 3c. In summary, we are convinced that indeed GTP is used by C12orf29.

We attempted to measure the ligase-GMP complex by MS (under the same conditions as used for the ligase-AMP intermediate) but were not able to detect it when using the very same GTP stock and only detected the unmodified (non-AMP-modified) ligase. This does not necessarily mean that the ligase-GMP intermediate was not formed. For example, it is possible that the intermediate is too unstable for MS analysis.

We have summarized these investigations in a “document for the reviewers” and refrained from including this in the original manuscript, since we feel that it would be rather distracting. Yet, if the reviewer wants to have this data included in the manuscript, we will be happy to do so.

4) It would be valuable to be more expansive in the Discussion on the coexistence of RtcB and C12orf29 type ligase in human/vertebrate cells. It is important to note that RtcB is essential for cell viability, because it is the agent of tRNA splicing, and is also essential for mRNA splicing in the unfolded protein response. By contrast, the present study makes clear that C12orf29 is not required for cell viability or (one surmises) for any essential RNA transaction when cells are grown in culture under standard conditions.

We agree with the reviewer and included a more thorough discussion on the coexistence of RtcB and C12orf29 in the discussion section (see 2nd last chapter starting with “While C12orf29 is probably not appear to be necessary...”).

5) I think the case for C12orf29 as a therapeutic target is weak and the text on this point ought to be deleted from the Abstract and the Discussion. The paper is interesting enough without such embellishment.

We agree and have revised the manuscript accordingly.

6) Introduction, lines 1-3. “mRNA alternative splicing” does not involve an RNA ligase. (The spliceosome catalyzes transesterification and is usually not considered to be a ligase, in that it does not synthesize a phosphodiester bond.) “rRNA maturation” does not involve an RNA ligase in most taxa. What is the evidence that an RNA ligase seals irradiation-induced RNA damage? These sentences (and citations) need reconsideration.

We have revised this part (very first sentence of the manuscript) and hope to have clarified this point.

7) p. 3. The authors briefly discuss the phylogenetic distribution of C12orf29 in the Introduction, with cursory reference to presence in “higher eukaryotes like vertebrates” and absence in “lower eukaryotes such as yeast”. The high/lower distinction is vague here as other arguably “higher” eukarya (i.e., metazoa) lack this protein. The phylogeny deserves more specific mention, either in Intro or in Discussion. The key points are that orthologs of C12orf29 protein are found in mammals, birds, reptiles, amphibians, and fishes (vertebrates) . . . and in a few invertebrates (mollusks). It is not present in not found in insects, arachnids, crustaceans, corals, worms, jellyfishes, sponges (invertebrate metazoa) or fungi.

We have added the phylogenetic information of C12orf29 accordingly in the introduction section of the revised manuscript (see Introduction part starting with “Sequence analysis shows that it is highly conserved ...”).

Text edits:

a) Do not start sentences with subjective opinion adverbs (unrelated to sentence verb).

-delete “Interestingly” (p. 4 and p. 6); “Intriguingly” (p5)

b) p. 3, line 16: delete “including humans” (redundant, as included in “higher eukaryotes”)

c) p. 3, lines 33-34: should be “highly conserved among vertebrates but absent in yeast” (delete “higher eukaryotes like” and “lower eukaryotes species such as”)

d) p. 3, line 28: delete “increasingly”

e) p. 5, line 17: should be “patterns . . . overlap” (not overlaps)

The text has been edited accordingly.

Reviewer #3 (Remarks to the Author):

The manuscript by Yuan et al reports an important novel enzyme function of the previously uncharacterized protein C12orf29 as the first known RNA ligase in human cells. This groundbreaking discovery is based on a chemical proteomic profiling with a C2-Ap3A probe (dimeric ATP with alkyne modification) which was expected to be used as a surrogate for ATP and transferred onto proteins. C12orf29 appeared as hit and was first confirmed to be modified with AMP. Further in-depth studies revealed a structural homology (based on alpha fold) to a RNA ligase (NgrRnl) and corresponding RNA ligation assays confirmed this notion. Moreover, in order to decipher the cellular role of this RNA ligase the authors made a knockout strain which was significantly more susceptible to oxidative stress resulting in enhanced degradation of 28S RNA further validating its crucial role for RNA integrity. The paper is well written and contains reports an exciting new enzyme functionality.

We are very pleased with this overall positive evaluation.

The following points require some attention:

1. The probe design and proteomic results need improved clarity. The authors mention in the discussion that their probes do not require external AMPylators for transfer and are thus superior to other methods. Recent methods with cell-permeable AMP prodrugs should be mentioned here as well as they utilize the cellular machinery in situ and also overcome these limitations.

We have added a discussion on this topic in the introduction of this manuscript (see: Several ATP derivatives have been designed for profiling AMPylation²⁰⁻²². In recent studies masked AMP analogues were used as probes to investigate AMPylation^{23,24}. After internalisation they were processed to ATP and used by the cellular machinery for protein AMPylation.).

In addition, the manuscript does not report any proteomic results and does not provide a rationale for the selection of C12orf29. Corresponding MS data, e.g. volcano plots of the enrichment should be shown at least in the SI and the raw data deposited at a public repository. What other proteins are identified with the C2-Ap3A probe and are any known AMPylators among the hits? This information would be important to better understand the selection of C12orf29 as well as the overall performance of the probe.

As written already above in response to reviewers 1 and 2, this assessment is equal to those of the other two reviewers.

We apologize for not having included the proteomics data as part of the originally submitted manuscript. We now list all significantly enriched interactors in a separate SI-table (as Excel

file, Supplementary Data 4) and have added paired comparisons in the Supporting Information (Supplementary Fig. 2). We have also deposited the raw files of the complete experimental dataset on the open access platform PRIDE. However, we decided against a lengthy description of our proteomics study in the main part of the manuscript, as we do not want to shift the focus from what we consider to be the most exciting part of this work, i.e. the discovery of a new RNA ligase in human cells.

C12orf29 was one of the completely uncharacterized proteins we identified in the proteomics data. This sparked our interest in deciphering its function. To reflect the motivation, we write in the manuscript: “Among the proteins identified from both human non-small cell lung carcinoma cells (H1299) and human embryonic kidney cells (HEK293T), one caught our attention since it had not yet been characterised: C12orf29 (see details in Supplementary Information and Supplementary Fig. 2).”

2. The authors performed kinetic experiments with C12orf29 and determine k_{cat} and K_M values. It would be interesting to compare these values with other RNA ligases. Is the performance in a similar range and if it is different, it would be interesting to discuss possible reasons?

There is only a very limited number of studies reporting on the kinetics of RNA ligases such as determination of K_M values. In fact, we found after intensive literature search only one study that determined a K_M of about 12 μM for ATP for T4 RNA ligase (see JBC 1974, vol 249, 7447); however, they used a very different RNA substrate (polyA) than we did in our study. Thus, we think that our experimental setup is very different to the one published and therefore refrained from a comparison.

REVIEWERS' COMMENTS

Reviewer #1 (Remarks to the Author):

This revised manuscript describes a discovery of an RNA ligase from chemoproteomic profiling. Based on this reviewer's comments, the authors revised manuscript to address all concerns properly, and this reviewer agrees that this material is ready for the publication. This is very interesting study and will be appealing to the community. Therefore, this reviewer suggests accepting this manuscript in this form in Nature Communications.

Reviewer #2 (Remarks to the Author):

The authors have addressed the major concerns in their revised manuscript.

A variety of text revisions need to be made prior to acceptance.

- 1) Abstract, line 2: "prokaryotes, and few of eukaryotes" is both awkward and inaccurate. "Prokaryotes" include bacteria and archaea. RtcB-type 3'-P/5'-OH ligases are more "prevalent" in these taxa than 5'-P/3'-OH ligases (all archaea and many bacteria have RtcB). If prevalent is the descriptor, then it is more accurate to say that 5'-P/3'-OH ligase is "prevalent in viruses, fungi, and plants." Note that 5'-P/3'-OH ligases are present in all known fungi and plants (these being eukaryal), of which there are many, not few.
- 2) Abstract, line 7: delete "the"
- 3) Abstract, line 1: better to say "establishing a human RNA repair pathway"
- 4) P. 3, line 13: delete the phrase "are widely prevalent ... some eukaryotes, which"
- 5) P. 7, line 9: should be "retained ~12% activity"
- 6) P 8, line 20-21: the authors should refer to the 3'-P/5'-OH ligase as RtcB (its proper name) not HSPC117. Moreover, it is false to state that RtcB "ligates such RNA termini in one step." RtcB catalyzes a multistep reaction pathway via covalent RtcB-His-GMP and RNAppG intermediates. The authors should explicitly cite Chakravarty et al. (2012 PNAS 109:6072) and Tanaka et al (2011 JBC 286: 43134) to that effect.
- 7) P 8, line 22; again re-write "viruses, prokaryotes and some eukaryotes" to indicate viruses, fungi and plants.
- 8) P 9, line 6; E. coli RtcB should be named as such, not as the HSPC117 homolog. The name RtcB predates HSPC117 by many years.

Reviewer #3 (Remarks to the Author):

My concerns have been addressed. Very nice paper!

Point-by-Point response to the Reviewer Comments

Reviewer #1 (Remarks to the Author):

This revised manuscript describes a discovery of an RNA ligase from chemoproteomic profiling. Based on this reviewer's comments, the authors revised manuscript to address all concerns properly, and this reviewer agrees that this material is ready for the publication. This is very interesting study and will be appealing to the community. Therefore, this reviewer suggests accepting this manuscript in this form in Nature Communications.

We are very pleased that our revision met the reviewer's satisfaction. We are very pleased with the reviewer's suggestion of acceptance.

Reviewer #2 (Remarks to the Author):

The authors have addressed the major concerns in their revised manuscript.

We are pleased that our revision met the reviewer's satisfaction.

A variety of text revisions need to be made prior to acceptance.

In the revision we have carefully addressed all points. This is described below.

Detailed comments:

1. Abstract, line 2: "prokaryotes, and few of eukaryotes" is both awkward and inaccurate. "Prokaryotes" include bacteria and archaea. RtcB-type 3-P/5'-OH ligases are more "prevalent" in these taxa than 5'-P/3'-OH ligases (all archaea and many bacteria have RtcB). If prevalent is the descriptor, then it is more accurate to say that 5'-P/3'-OH ligase is "prevalent in viruses, fungi, and plants." Note that 5'-P/3'-OH ligases are present in all known fungi and plants (these being eukaryal), of which there are many, not few.

We agree with the reviewer and are grateful for this comment. We have amended the text that meets the requirement by saying "While enzymatic RNA ligation between 5'-PO₄ and 3'-OH termini is prevalent in viruses, fungi, and plants, such RNA ligases are yet to be identified in vertebrates."

2. Abstract, line 7: delete "the".

We have amended the text that meets the requirement.

3. Abstract, line 1: better to say “establishing a human RNA repair pathway”.

We have amended the text that meets the requirement.

4. P. 3, line 13: delete the phrase “are widely prevalent ... some eukaryotes, which”.

We have amended the text that meets the requirement.

5. P. 7, line 9: should be “retained ~12% activity”

We have amended the text that meets the requirement.

6. P 8, line 20-21: the authors should refer to the 3'-P/5'-OH ligase as RtcB (its proper name) not HSPC117. Moreover, it is false to state that RtcB “ligates such RNA termini in one step.” RtcB catalyzes a multistep reaction pathway via covalent RtcB–His-GMP and RNAppG intermediates. The authors should explicitly cite Chakravarty et al. (2012 PNAS 109:6072) and Tanaka et al (2011 JBC 286: 43134) to that effect.

We agree with the reviewer and are grateful for this comment. We have amended the text that meets the requirement by saying “So far, only a GTP-dependent 3'-5' RNA ligase, RtcB (or HSPC117)⁹, has been reported in human, which ligates such RNA termini via successive auto- and RNA-GMPylation^{43,44}.” The two papers mentioned by the reviewer have been cited accordingly as references 43 and 44.

7. P 8, line 22; again re-write “viruses, prokaryotes and some eukaryotes” to indicate viruses, fungi and plants.

We have amended the text that meets the requirement.

8. P 9, line 6; *E. coli* RtcB should be named as such, not as the HSPC117 homolog. The name RtcB predates HSPC117 by many years.

We agree with the reviewer and are grateful for this comment. We have amended the text that meets the requirement. We have changed the text as following:

“In contrast, the RNA ligase RtcB and its homologs are reported to play vital roles in intron-containing tRNA maturation⁹ and *X-box binding protein 1 (XBP1)* mRNA splicing during UPR³. Furthermore, it has been suggested that the *E. coli* RtcB can repair nuclease- or antibiotic-induced rRNA damage^{49,50}. The recently identified 2',3'-cyclic phosphatase ANGEL2 can hydrolyse the 2',3'-cPO₄ at 3'-termini to 2'-OH-3'-OH, thereby antagonising RtcB-mediated tRNA and *XBP1* mRNA splicing³⁶.....Therefore, we speculate that C12orf29 may be involved in

an RNA ligation machinery in concert with other RNA processing enzymes when the RNA termini and activities are compromised in RtcB-mediated RNA ligation system.”

Reviewer #3 (Remarks to the Author):

My concerns have been addressed. Very nice paper!

We are very pleased that our revision met the reviewer's satisfaction.